# Cancer Vaccines: Molecular Mechanisms, Clinical Progress, and Combination Immunotherapies with a Focus on Hepatocellular Carcinoma

**DOI:** 10.3390/cimb47121056

**Published:** 2025-12-17

**Authors:** Faris Alrumaihi, Reem A. Alromaihi, Vikalp Kumar, Shehwaz Anwar

**Affiliations:** 1Department of Medical Laboratories, College of Applied Medical Sciences, Qassim University, Buraydah 51452, Saudi Arabia; f_alrumaihi@qu.edu.sa (F.A.);; 2Department of Medical Laboratory Technology, Mohan Institute of Nursing and Paramedical Sciences, Bareilly 243302, Uttar Pradesh, India

**Keywords:** hepatocellular carcinoma, cancer vaccines, tumor-associated antigens, immunotherapy

## Abstract

Conventional cancer treatments often fail due to the immunosuppressive tumor microenvironment, immune tolerance, and chronic inflammation. Therefore, new therapeutic approaches are urgently needed. Cancer vaccines can stimulate natural killer cells and cytotoxic T-lymphocytes, and induce long-lasting memory responses that help overcome the immunosuppressive tumor microenvironment. Recent advances in nucleic acid, peptide, and dendritic cell-based vaccines have improved antigen delivery and immune activation, while combinations with immune checkpoint inhibitors and ablative therapies enhance therapeutic efficacy and durability. Preclinical and clinical studies targeting tumor-associated antigens have shown promising outcomes. With poor survival rates and limited treatment options, hepatocellular carcinoma (HCC) appears to be the most prevalent cause of cancer-related deaths worldwide. Advances in antigen discovery, vaccine delivery systems, and synergistic combination strategies are paving the way for more effective and durable immune responses. By integrating molecular insights with clinical innovation, cancer vaccines hold the potential not only to improve treatment outcomes but also to redefine long-term disease management and survival in HCC.

## 1. Introduction

Cancer remains one of the leading global health challenges, accounting for nearly 10 million deaths worldwide. It is characterized by uncontrolled proliferation and spread of abnormal cells that invade nearby tissues and migrate to distant organs. Recent studies have revealed the complex interactions among cancer cells, stromal cells, immune cells, and other components of the extracellular matrix within the tumor microenvironment (TME), all of which affect the course of the tumor and its response to treatment [1]. Both exogenous factors, including nutrition, the environment, nicotine, chemicals, lifestyle, obesity, diet, and infectious organisms, as well as intrinsic factors, like inherited mutations, genetic predisposition, and immunological disorders, play a critical role in carcinogenesis [2].

Recent technological advancements in multi-omics platforms have led to a deeper understanding of tumor heterogeneity. However, treatment failure is a significant problem for effective cancer treatment, and this failure can be attributed to both intra-tumoral and inter-patient heterogeneity. In this regard, early cancer identification can reduce the risk of dying from cancer, screen for probable cancers before symptoms appear, and have more potential benefits than drawbacks [3].

Cancer patients often experience poor outcomes due to metastasis, recurrence, and treatment-related toxicities, even with advances in traditional therapies such as radiation, chemotherapy, and surgery [4]. Immunotherapy has transformed cancer care by harnessing the body’s immune system to recognize and eliminate cancer cells. Extensive research in this field has led to the approval of several innovative modalities, including immune checkpoint inhibitors (ICIs), cancer vaccines, and chimeric antigen receptor (CAR)-T and T-cell receptor (TCR) therapies [5]. These new approaches offer promising strategies for mobilizing the host immune system against a wide range of cancers [6,7].

In immunotherapeutic approaches, foreign or newly produced proteins serve as potential targets. Specifically, these neoantigens facilitate the activation of CD8^+^ T cells to directly contact and kill cancer cells [8]. Consequently, cancer vaccines represent a significant approach for stimulating robust, antigen-specific adaptive immune responses against tumors [9].

In contrast to traditional immunizations, which protect against infectious diseases, cancer vaccines are made to detect and destroy cancer cells. They can be used as prophylactic vaccinations to prevent cancer in people who are at high risk of acquiring certain forms of cancer, or as therapeutic vaccines to treat cancer that already exists. Cancer vaccines, which can be used for both cancer prevention and treatment, are therefore a promising form of cancer immunotherapy. They could be a potent tool in the fight against cancer [10].

Therapeutic cancer vaccines harness the immune system to selectively recognize and eliminate tumor cells by targeting tumor-associated or tumor-specific antigens. Unlike preventive vaccines, therapeutic vaccines are designed to treat existing malignancies by enhancing immune surveillance and triggering robust anti-tumor immunity [8].

Despite their potential, several challenges hinder their clinical efficacy, including inadequate targeting of delivery systems, vaccine stability and safety, insufficient immunogenicity, and the immunosuppressive tumor microenvironment. To overcome these challenges, numerous advanced delivery platforms such as liposomes, outer membrane vesicles (OMVs), and nanoparticles have been developed. These platforms may offer improved stability, antigen presentation, and targeted administration with controlled release [5].

Recent advances in immunology and molecular biology have enabled more sophisticated vaccines platforms, including viral, cellular, peptide-based, and nucleic acid approaches. Building on these innovations in immune regulation, antigen design, and delivery technologies, the present review examines the growing significance of cancer vaccines in the treatment of hepatocellular carcinoma (HCC). We summarize recent progress in nucleic acid, peptide, and cell-based vaccine strategies and discuss their potential integration with ablative therapies and ICIs to improve therapeutic efficacy and durability. By evaluating current preclinical and clinical evidence, this review highlights the translational potential of cancer vaccines as a promising immunotherapeutic modality for improving outcomes in patients with HCC.

## 2. Cancer Vaccines

Cancer vaccines are developed to enhance immune recognition of tumor antigens and promote the selective elimination of cancer cells by targeting tumor-associated antigens (TAAs) [11]. Cancer vaccines may contain dendritic cells (DCs), tumor antigens, whole tumor cells or lysates, or nucleic acids. Among these, DC-based vaccines are the most extensively studied and are known to enhance antigen presentation and activate tumor-specific lymphocytes. These vaccines are typically administered subcutaneously, intramuscularly, or intravenously, depending on the formulation and treatment strategy. Cancer vaccines work through various mechanisms, as shown in Figure 1 [12].

Once introduced into the body, antigen-presenting cells (APCs), particularly dendritic cells, phagocytose or express tumor antigens, and then efficiently process them intracellularly and display them on their surface via major histocompatibility complex (MHC) molecules. By binding to T-cell receptors (TCRs) on the surface of T cells, MHC–antigen complexes trigger the activation of antigen-specific cytotoxic T cells, which selectively eliminate tumor cells and prevent tumor growth [13].

Cancer immunotherapy encompasses vaccines, immune check inhibitors (ICIs), adoptive cell transfer (ACT), oncolytic viral therapy, antibody-based therapies, and cytokine therapies. By targeting molecules such as cytotoxic T-lymphocyte-associated antigen 4 (CTLA-4) and programmed death 1/programmed cell death ligand-1 (PD-1/PD-L1), ICIs enhance the immune system’s ability to identify and aggressively combat cancer cells. Cytokine-based therapies, including interleukins and interferons support immune cell signaling and can augment antitumor responses [14].

Generally speaking, tumor immunotherapy can be broadly categorized into passive (or adaptive), which involves administering cells or antibodies outside of the body, and active, which includes cancer vaccines that stimulate endogenous immune response against tumor-specific antigens (TSAs) and tumor-associated antigens (TAAs). Preventative and therapeutic vaccines remain particularly promising, as vaccine-induced immunity has potential to generate a durable, long-lasting protection compared with conventional treatment [15].

### 2.1. Nucleic Acid-Based Vaccines

Nucleic acid vaccines, such as DNA and mRNA vaccines, represent a novel form of cancer immunotherapy (Figure 2).

Nucleic acid vaccines stimulate the immune system by delivering genetic material that encodes TAAs or neoantigens. The genetic material in a DNA vaccine is delivered into the cell nucleus, where it is transcribed into mRNA, which is then translated into the target protein. For mRNA vaccines to initiate translation of the target protein, they must enter the cytoplasm. Because DNA vaccines require an additional step to enter the cell nucleus, mRNA vaccines are often more effective at generating antigens. But compared to mRNA vaccines, DNA vaccines are more stable and retain their effect longer. Furthermore, other mRNAs can be produced in the cytoplasm once the DNA vaccine has effectively entered the cell nucleus [8].

#### 2.1.1. DNA Vaccines

DNA vaccines represent a straightforward platform for inducing antigen expression and generating cells in vivo. They typically consist of plasmid DNA, produced in bacteria. It encodes the target antigen under the control of a mammalian promoter such as the CMV-intA, CMV immediate/early promoter, and its neighboring intron A sequence. Recently, DNA-based vaccines have gained attention for their ability to generate immunity against weak TAAs and to deliver single or multiple tumor antigens or to modify antitumor immune responses [16,17,18].

Vaccine efficacy can be significantly increased through improved antigen presentation and immunogenicity, including optimized delivery systems, molecular adjuvants, immunostimulatory signals, prime-boost optimization, or immune checkpoint blockade. Through these approaches, DNA vaccines have shown the ability to reduce tumor growth and to induce complete tumor rejection by activating both innate and adaptive immune responses [19].

Plasmid DNA molecules encoding TAAs or TSAs, linked to the targeted tumor under the direction of a mammalian promoter, comprise DNA cancer vaccines [20]. The host’s molecular machinery expresses TAAs or TSAs following transfection of the host cell and plasmid transfer into the transfected cell’s nucleus. Following processing, MHC proteins bind immunogenic epitopes, which in turn trigger the CD4^+^ and CD8^+^ T cells and the humoral and cellular immune responses that accompany them. There are several ways to give DNA vaccines. Still, the most common are intradermal and intramuscular APCs, which primarily use MHC-I to present endogenous antigen to CD8^+^ cells when directly transfected. However, some antigens will also be cross-presented to CD4^+^ T cells via MHC-II, which predominantly triggers the humoral immune response [21,22].

Through a variety of processes, including phagocytosis and apoptosis induction, the stimulated immune cells may be able to identify the tumor cells and eliminate tumor cells. The primary obstacle to the widespread use of DNA vaccines is their very low immunogenicity. Cancer cells use a variety of tactics to evade immune system detection [23].

DNA vaccines can also activate an immune response through uptake of apoptotic bodies or exosomes released by transfected myocytes or keratinocytes. DCs internalize these vesicles and present antigen via MHC II to CD4^+^ T cells, whereas direct transfection of antigen presenting cells (APCs) induces endogenous antigen expression and simultaneous CD8^+^ and CD4^+^ activation. The B cell may also recognize antigens released from transfected somatic cells, including myocytes and keratinocytes, in concert with the cellular immune response. Pre-activated, antigen-specific CD4^+^ T cells that can start a humoral immune response help in this detection [24,25].

Several strategies have been developed and tested to increase the effectiveness of DNA-based vaccines, such as the use of distinct plasmids encoding non-self-antigens (e.g., hepatitis B surface antigen), simultaneous administration of cytokines (GM-CSF or IL2), and improved delivery systems (Gene Gun, cationic liposomes). Different alterations to plasmid-encoded antigens can also increase the immunogenicity of DNA-based vaccinations [26].

Sushack and colleagues highlight that DNA vaccines offer many advantages, including safety, low cost, ease of manufacturing and storage, and the ability to incorporate multiple antigens or immune-stimulating components. They promote sustained antigen expression, induce both humoral and cellular immunity, and can be administrated repeatedly without vector-neutralization issues. However, significant drawbacks include long production times required for personalized vaccine development, relatively low immunogenicity, poor responses, and risks such as insertional mutagenesis or horizontal transfer of antibiotic resistance genes. Techniques such as electroporation and sonoporation have been used to improve plasmid delivery and enhance vaccine efficacy [27].

Electroporation remains one of the most effective techniques, temporarily increasing cell membrane permeability to facilitate plasmid entry. However, each transfected can activate only a limited number of immune cells, since the encoding DNA is not readily transferred from cell to cell in vivo. Second, TAAs’ immunogenicity is limited by both central and peripheral mechanisms of tolerance because they are self-antigens [25]. Table 1 overviews DNA vaccines trials across multiple cancer types, antigen targets, administration schedules, and safety outcomes.

#### 2.1.2. RNA Vaccines

RNA-based cancer vaccines have emerged as an important class capable of stimulating the immune system to identify and target specific cancer cells [26]. Because RNA is translated directly in the cytoplasm without entering the nucleus, these vaccines rapidly generate antigen expression while avoiding the risk of genome integration or T cell tolerance. By encoding full-length tumor antigens, RNA vaccines allow APCs to process and present a broad spectrum of epitopes via HLA class I and II molecules. As a result, they are more likely to elicit a wider variety of T-cell responses and are less type-restricted by human HLA. Integration into the host genome is not a risk [28].

Wang et al. reported that mRNA cancer vaccines exhibit strong immunogenicity, inducing robust humoral and cell-mediated immune responses that contribute to a potent antitumor effect. Owing to their ability to trigger systemic immunity, mRNA cancer vaccines have demonstrated promise against metastatic tumors that are challenging to treat surgically. Moreover, they can generate durable immunological memory, supporting long-term protection and reducing the likelihood of tumor recurrence. Multiple preclinical cancer studies in both primary tumors and metastases have demonstrated the strong therapeutic efficacy of mRNA vaccines [29]. Conventional and self-amplifying mRNAs are the two types of mRNA that are utilized as vaccines. With their 5′ cap, 5′ UTR, coding region, 3′ UTR, and polyadenylated tail, conventional mRNAs resemble endogenous mRNAs seen in mammalian cells [30,31].

Cellular absorption of RNA is hindered by its hydrophilicity and large net negative charge. To overcome this restriction, physical delivery methods such as ballistic particles or electroporation have been studied to enhance cellular uptake, as have electrostatic complexation with cationic lipids or polymers. This kind of mRNA is translated in the cytosol and then broken down without further replication. However, the genomes of single-stranded RNA viruses, including alpha viruses, serve as a source of self-amplifying mRNAs. Non-amplifying mRNA vaccines can be manufactured in sufficient quantities and with high quality to meet regulatory standards, and they are currently being studied in human clinical trials. This platform could establish nucleic acid vaccines as a flexible new tool for human vaccination if the promising preclinical findings with self-amplifying mRNA vaccines are matched by similarly favorable immunogenicity, potency, and tolerability in human trials [32].

Personalized medicines can boost therapeutic efficacy and reduce adverse effects, another critical benefit of mRNA cancer vaccines [33,34]. Highly adaptable mRNA vaccines make it easy to modify mRNA sequences to encode specific antigens or cytokines useful in the fight against cancer [33,35]. The mRNA cancer vaccine is a nucleic acid vaccine that does not require nuclear entry and can be translated as soon as it enters the cytoplasm [36]. There is no risk of unintended infection with mRNA cancer vaccines, which makes it a crucial safety concern [37,38]. mRNA cancer vaccines may circumvent the need for nuclear localization and the risk of insertional mutagenesis associated with DNA vaccines [39].

Sample collection, gene sequencing and target identification, mRNA sequence design, vaccine manufacturing, and delivery of the finished vaccine product are all steps in the development and manufacturing process of mRNA cancer vaccines. Internalization of the vaccine into cells, transcription of mRNA sequences encoding specific targets, distribution to immune cells, immune system stimulation, and tumor cell death are all components of the mechanism of action of mRNA cancer vaccines [29]. Table 2 provides an overview of clinical evaluation of mRNA-based cancer vaccines across multiple tumor types.

### 2.2. Peptide-Based Cancer Vaccines

The potential of peptide-based therapeutic cancer vaccines to enhance patient outcomes has attracted significant interest. The U.S. Food and Drug Administration (FDA) has only authorized one peptide-based cancer vaccine. To advance these vaccines, a thorough grasp of the underlying mechanisms and present state of development is essential [39]. In this type of vaccination, the unique 8–12 aa peptide from the tumor antigen (TA) coding sequence is used. Proteins that are overexpressed and emerge during carcinogenesis and development give rise to tumor antigens (TAs). For T-cell activation, it may be internalized into DCs, where it is broken down into peptides and assembled onto the surface of human leukocyte antigen (HLA) molecules. T cells can identify the unique peptide-HLA complex in addition to specific TA [40].

The amino acid sequence used in peptide-based cancer vaccines is usually derived from tumor-specific or tumor-associated antigens (TSA/TAA). For peptide-based cancer vaccines to be effective, they need both CD8^+^ epitopes, which can exploit antigen cross-presentation pathway and activate CTL antitumor immunity, and CD4^+^ epitopes, which can activate T-helper cells and maintain CTL effector activity [41]. Therefore, the sequence length of peptide vaccines is crucial for promoting a robust immunogenic response. Because non-professional APCs lack the secondary signaling apparatus necessary for full T-cell activation, a peptide that is too short may bind to their MHC, leading to a weak T-cell response or immunological tolerance [42].

Shorter peptides do not provide the diversity required by the high polymorphism of HLA in the general population [43,44]. They are therefore more likely to be restricted by their HLA type. Subsequently, short peptides are more likely to be broken down by enzymes and eliminated from the body faster if they are not modified. However, a longer peptide length enables the addition of multi-epitope peptides to enhance CD4^+^ and CD8^+^ responses, incorporation of binding or recognition motifs to increase immunogenicity, and greater population coverage of HLA types [45].

Amino acid sequences are efficiently assembled by peptide synthesis, usually solid-phase peptide synthesis. Following purification to remove impurity, typically using high-performance liquid chromatography (HPLC), the peptides are characterized using techniques such as analytical HPLC and mass spectrometry to verify their identity and purity [46]. The peptides are made with adjuvants to boost the immunological response. Adjuvants modify the specificity of the immune response by altering the peptide antigens to which CD4^+^ T cells are exposed [47].

Aluminum salts, incomplete Freund’s adjuvant (IFA), and MF59 are commonly used adjuvants selected for their ability to improve the immune response by activating the innate immune system and enhancing antigen presentation [48]. To prevent peptide degradation, ensure targeted distribution, and encourage uptake by antigen-presenting cells, delivery technologies such as liposomes, nanoparticles, and emulsions are used. Peptide stability must also be addressed in the formulation, with conditions optimized to avoid degradation. Numerous factors, such as dose, route of administration, and dosing schedule, affect immunogenicity [39]. The final product is usually given as an injection, either intradermally, intramuscularly, or subcutaneously, depending on the specific vaccine design. It can be prepared in a liquid form for an instant infusion or in lyophilized form that requires reconstitution prior to administration [49,50].

Peptide vaccines function by delivering short, synthetic peptide fragments that mimic tumor-associated or tumor-specific antigens. Once administered, these peptides are taken up by APCs, such as dendritic cells (DCs), and processed and presented on major histocompatibility complex (MHC) molecules. Peptides bound to MHC-I molecules activate cytotoxic CD8^+^ T lymphocytes, enabling direct recognition and killing of tumor cells that display the same antigen. At the same time, peptides presented on MHC-II molecules stimulate CD4^+^ helper T cells, enhancing and sustaining cytotoxic T cell and B cell responses. This coordinated activation of adaptive immunity provides a promising approach to cancer immunotherapy, producing a targeted and durable antitumor effect [39].

Peptide vaccines offer several advantages over conventional therapies. They are safer than chemotherapy and radiotherapy, with minimal toxicity, and unlike CAR-T cell therapy, they can target the surface and intracellular tumor epitopes. By excluding B-cell epitopes, they reduce the risk of hypersensitivity and effectively address tumor heterogeneity. These vaccines can overcome antigen loss by combining numerous epitopes, enhancing flexibility, and elicit robust immune responses for active immunotherapy. Furthermore, individualized neoantigen peptide vaccines have demonstrated safety, efficacy, and strong T-cell activation, and genetic changes in neoantigens can elicit natural T-cell responses [51]. Table 3 summarizes key clinical trials of peptide-based cancer vaccines across multiple cancer types.

### 2.3. Tumor Cell-Derived Vaccine

New approaches that promote therapeutic strategies are highly desirable because, despite conventional cancer therapies, neoplasm metastasis and cancer recurrence make cancer treatment a medical challenge. One way to address these challenges as a novel treatment is to employ whole neoplastic stem cells or vaccines targeting cancer stem cells (CSCs) [52].

Therapeutic vaccines with well-defined antigens have been used to treat cancer. However, their extensive clinical applicability is limited by patient antigenic heterogeneity and high cost. Additionally, insufficient production of target proteins, inability to achieve adequate local concentrations, or insufficient exposure duration are common issues with vaccines made from inactivated tumor cells. A strong immune response might be elicited by developing a whole tumor cell vaccine that contains all tumor cell antigens, along with their relevant conformations. This method is effective for treating advanced malignant tumors. Depending on the source of the cells used to produce the vaccine, TCVs can be divided into three categories: autologous whole TCV (ATCVs), allogeneic-derived whole TCV, and cancer stem cell (CSC) vaccines [53].

For over two decades, researchers have examined the effectiveness of whole-cell tumor vaccines in preclinical models and human clinical trials. Whole-cell/polyepitope vaccines have distinct advantages over immunotherapies that target individual epitopes. A variety of unknown antigens may target both innate and adaptive immune systems, and this effect may be further enhanced by genetically modifying the vaccination cells to produce costimulatory signals and cytokines [54].

It is essential to prepare intact tumor cell lysates that induce cell death and prevent the release of tumor-derived cytokines that hinder vaccine-driven immune activation. In general, promising immunotherapeutic strategies include modified whole tumor cells, tumor cell-derived exosomes, autologous tumor cell-derived ribonucleic acid, and customized mutanome-derived tumor antigen. As a successful cancer therapy strategy, autologous dendritic cells loaded with TAAs also stimulate the development of immunological memory and an anticancer response [55]. To replicate the natural immune system and induce an adaptive immune response against tumor antigens, cell-based cancer vaccines employ autologous patient-derived tumor cells, allogeneic cancer cell lines, or autologous APCs [56].

This includes vaccines made of tumor cells and dendritic cells, which are undoubtedly different [57]. Allogeneic and autologous tumor cells are the two methods available for cell-based vaccinations. Despite not being personalized, allogeneic vaccinations have the benefit of saving time. Autologous vaccines, on the other hand, use the patient’s own tumor cells, ensuring antigen compatibility but at the penalty of higher expenses and longer turnaround times. Several variables, such as the patient’s specific requirements, the stage of the cancer, and the resource availability, influence the choice of methods. The possibility of human leukocyte antigen (HLA) mismatch is a significant disadvantage of cell-based vaccinations. This discrepancy may shift the focus from the immune system to HLA molecules, thereby jeopardizing the efficacy of the vaccine [11]. Table 4 summarizes cell-based cancer vaccines across various cancer types.

#### 2.3.1. Whole-Cell-Based Cancer Vaccines

In this technique, whole-cell preparations are derived from patient tumors or known cancer cell lines, and the cells are either inactivated or genetically modified to remove their tumor-causing potential. Upon reintroduction into the patient, they are recognized by dendritic cells, macrophages, and natural killer cells, which initiate an immediate inflammatory response. The immune system processes and presents TAAs to T cells, thereby activating CD4^+^ helper T cells to support immune signaling and CD8^+^ cytotoxic T lymphocytes to eliminate tumor cells directly. Importantly, whole cell vaccines are also designed to induce long-term immunological memory, providing sustained protection against tumor recurrence [11].

As the most potent professional APCs, DCs play a central role in initiating adaptive antitumor immunity antigens to naive CD4^+^ and CD8^+^ T cells. A key mechanism is cross-priming, in which DCs present exogenous tumor antigens to activate CD8+ T cells via MHC I molecules. Effective antitumor immunity, therefore, relies on efficient antigen uptake, cross-presentation, and subsequent effector CTL-mediated tumor cell killing. However, tumor-infiltrating DC (TIDC) within TME often become functionally suppressed, resulting in impaired anti-tumor CD8^+^ T-cell activation and reduced cross-priming efficiency [58].

The primary strategy involves isolating DCs from the patient’s blood or generating them ex vivo. These DCs are then matured and stimulated with immune adjuvants or tumor adjuvants. After reinfusion with TSAs obtained from tumor lysates or genetic material, they are reintroduced into the patient. Once administered, DCs migrate to lymphoid organs, where they interact with T cells, B cells, and NK cells. By presenting tumor antigens to T cells, DCs promote a robust cytotoxic and immune response. Activated CTL cells specifically recognize and destroy cancer cells that express tumor antigens. The memory T cells provide long-lasting immune-surveillance [11,59,60,61].

Several approaches have been explored to use DCs as therapeutic vaccines to harness their anti-tumor cell activity. The most common methods for pulsing DCs involve using whole tumor cells or purified/recombinant antigen peptides, followed by reinfusion into patients. Although preclinical and clinical trials have shown encouraging outcomes, limitations are linked with overall efficiency. The main challenges contributing to therapeutic failure include the presence of highly suppressive TMEs, upregulation of immune checkpoint inhibitors, low avidity of TAA-specific T cells, and the use of suboptimal adjuvants [61]. DC-based vaccines loaded ex vivo with tumor antigens have demonstrated strong immunogenicity and high response rates in clinical studies, suggesting promising potential for cancer immunotherapy [62].

To improve immunogenicity, complete tumor cells encoding cytokines such as IL-2, IL-12, and GM-CSF are prepared from a patient’s tumor cells [55]. Allogeneic vaccines prepared from established tumor cell lines from several members of the same race offer the logistical benefits of easy, standardized quality control and large-scale production. However, unmodified tumor cells often fail to induce a strong immune response [63].

Liquid nitrogen-based cryoshocking can produce non-proliferative tumor cells that retain structural integrity and can function as a tumor vaccine as well as drug-delivery carrier. Cryo-shocked AML cells retained their migratory potential in bone marrow and delivered doxorubicin (DOX) effectively. Unlike living cell systems, these cryoshocked delivery vehicles may be rapidly prepared for clinical use [64]. Cryoshocked tumor cells combined with adjuvants have been tested in lung cancer models. For example, Tadao et al. found that injecting an MS-Ap-PAMA adjuvant into C57BL/6 mice with liquid nitrogen-based Lewis lung carcinoma significantly reduced tumor recurrence and inhibited the tumor growth upon rechallenge [65].

GVAX cancer vaccine, consisting of GM-CSF-secreting genetically modified tumor cells, promotes DC activation [66]. However, preclinical studies demonstrated tumor progression and extended survival. The GVAX-PCa version, which consists of a mixture of irradiated PC-3 and LNCaP prostate cancer cell lines overexpressing GM-CSF, progressed to phase I/II trials, showing safety and induction of antigen-specific immune response [67].

Sipuleucel-T represents autologous cellular immunotherapy for metastatic castration-resistant prostate cancer. Patient APCs are activated ex vivo using PA2024 (PAP–GM-CSF fusion), and reinfusion stimulates targeted CTL responses against PAP-expressing tumor cells [68].

#### 2.3.2. Tumor Cell Lysate-Based Cancer Vaccine

The use of homologous tumor lysates provides a broad range of TAAs, enabling presentation through both MHC class I and II pathways, and generating a stronger and more diverse T-cell response than peptide-based vaccines. However, their immunogenicity is limited by a relatively low number of TSAs within TCLs, poor APC uptake, and rapid antigen degradation, prompting the development of engineering approaches to improve their stability, uptake, and immunogenicity [69].

Tumor tissue or cell lysate is a cheap and safe source of antigens that contains the entire repertoire of the patient’s TSAs (neoantigens). Repetitive rapid freeze–thawing of cancer cells produces a mixture of cellular components, including fragments of the disrupted cellular membrane, mitochondria, and cellular RNA and DNA. Moreover, toll-like receptors (TLRs) on APCs detect high mobility group protein (HMGB1), calreticulin, ATP, uric acid, nucleic acids, and lipids that are released when tumor cells undergo freeze–thaw lysis [64].

PAM-treated tumor lysates showed enhanced DC maturation, increased expression of CD83, CD86, and CD40, favorable cytokine profiles (IL-12/IL-10), and reduced expression of PDL1 and ILT-4. DCs loaded with these PAM-lysates also retained a strong Th1 response and increased the percentage of cytotoxic IFN-γ+ granzyme A+ CD8^+^ T cells and IL-17A-producing T cells when co-cultured with allogeneic T cells. Collectively, these findings imply that PAM-generated immunogenic tumor lysates may serve as a more immunogenic platform for enhancing DC-based cancer immunotherapy [70].

In one study, tumor cell lysate (TCL) antigens were covalently conjugated to polydopamine (PDA) via Michael addition and Schiff base reactions, linking the free and sulfhydryl groups in the lysate to the catechol groups of PDA. The resulting TCL@PDA nanoparticles enhanced the expression of surface costimulatory molecules, cytokine release, and antigen uptake by DCs, thereby effectively suppressing tumor growth. Whole cell lysis fractions contain both water-soluble and water-insoluble antigens. A broader range of tumor antigens is presented to the body when a water-insoluble adjuvant is included in a tumor vaccination [71].

Similarly, hypochlorite treatment has been shown to increase immunogenicity of TCLs in another study [72]. Berti et al. developed PLGA nanoparticles loaded with hypochlorite-oxidized tumor lysates, which significantly enhanced antigen uptake and induced DC maturation. As a result, animals immunized with PLGA nanoparticle formulation exhibited prolonged survival compared with those receiving the vaccine containing free oxidized tumor lysate [73]. In another approach, mannose-modified chitosan nanoparticles (Man-CTS NPs) were used as a carrier for TCL delivery, facilitating targeted uptake by DCs via mannose receptor-mediated endocytosis and enhancing the subsequent immune response [74].

### 2.4. Tumor-Derived Extracellular Vesicle-Based Vaccines (TEVs)

Extracellular vesicles, or EVs, are tiny membrane-bound particles secreted by almost every type of cell types. They play a central role in intercellular communication by transferring proteins, nucleic acids, and various lipids and can target specific tissues. Because of this natural ability, they are frequently employed as effective drug carriers. Exosomes, microvesicles, apoptotic bodies, and oncosomes are the four subcategories of EVs. EVs are widely explored as therapeutic tools for various diseases, including cancer. TEVs have a unique composition and the capacity to promote the maturation of immune cells. They can initiate an anticancer response, making them promising, potent tumor vaccines [75].

#### 2.4.1. Tumor Cell-Derived Exosome Vaccines

Tumor-derived exosomes (TEXs) are nano-sized vesicles that carry a range of useful molecular content originating from the cell membranes and nuclear endosomes of primary tumor cells. They have been identified in several malignancies, including cancer of the kidney, blood, breast, and skin, and can efficiently transport their content to recipient cells [76,77]. Owing to the proteins present on their membranes, they exhibit strong cellular uptake, enabling them to deliver internal molecular cargo to tumor sites and stimulate immune responses. After being internalized by DCs, antigens from donor tumor cells in TEXs are processed and presented via MHCs, leading to activation of naïve T cells and the induction of antitumor responses [78,79].

TEXs can evade the immune system’s attacks due to the presence of transmembrane CD47^+^ [80]. Moreover, cancer cells typically release ten times more TEXs than normal cells [81]. In one study, the MHC II transactivator protein CIITA (Class II transactivator) gene was transduced into melanoma B16F10 cells, resulting in the production of TEXs enriched with MHC class II molecules. Vaccination with these CIITA-TEXs significantly increased the mRNA levels of pro-inflammatory cytokines, such as TNF-α, and the chemokine receptor 7. It increased the surface expression of MHC class II and CD86 on DCs. Thereby, CIITA-TEXs induced a potent immunological response [82]. Similarly, Fan et al. demonstrated that CIITA-transduced TEXs derived from the rat colon cancer CT-26 cell line yielded comparable outcomes. Elevated TNF-α, IFN-γ, and IL-12, along with a reduced IL-10 expression, demonstrated that CT26-CIITA-derived TEXs enhanced Th1-type immune responses [83].

#### 2.4.2. Tumor-Derived Microvesicle-Based Vaccine

Tumors or other cells within TME release tumor-derived microvesicles (TMVs). They contain a variety of proteins, tumor-associated nucleic acids, and other bioactive substances that alter invasion, angiogenesis, immunological responses, and tumor metastasis. TMVs are released directly from the plasma membrane and can be stimulated by exposure to radiation and hypoxia [84]. Pineda et al. demonstrated a therapeutic tumor vaccine based on TMVs derived from C6 glioma cells following radiation exposure, leading to tumor cell death and enhanced T cell infiltration in vaccinated rat tumors, ultimately resulting in a significant reduction in tumor volume [85].

### 2.5. Virus-Based Cancer Vaccine

Inactivated viral vaccines are produced using viruses that have been destroyed or rendered non-infectious, so they can no longer spread disease. The immune system can identify the virus after vaccination, even in the absence of an active infection. Antigen-presenting cells (APCs), such as DCs, process viral particles and display viral antigens on their surfaces, activating T cells and B cells. This leads to the antibody production and memory cell formation. As a result, the immune system can respond more rapidly and efficiently upon future encounters with these cancer-causing viruses [86].

Oncolytic viral cancer vaccinations can selectively destroy cancer cells. These oncolytic viruses (OVs) replicate within cancer and related endothelial cells. Numerous OVs naturally exhibit a preference for tumor and/or related endothelial cells. However, other OVs have been engineered to enhance their specificity for cancer-related signaling alterations, including RB/E2F/p16, p53, PKR, EGFR, Ras, Wnt, anti-apoptosis, hypoxic conditions, and defects in interferon (IFN) signaling. Viral replication under these conditions promotes direct oncolysis, leading to tumor cell death and the release of tumor antigens, which further stimulate antitumor immunity [87].

The majority of viruses can be genetically engineered to express tumor antigen transgenes and are immunogenic by nature. Furthermore, these recombinant viruses can infect professional APCs, particularly DCs, facilitating enhanced antigen presentation and robust CTL responses against tumor cells expressing the encoded antigens. Such systems are comparatively easier to construct, administer, and scale, making them appealing for therapeutic use. The frequency and avidity of CTLs, which target tumor cells expressing the tumor antigen(s) encoded by the vaccine vector, have increased as a result of the improved presentation of tumor antigens to the immune system. Compared with other immunotherapy approaches, recombinant viruses are easier to create, administer, and manage logistically [88].

Viral vectors are among the most widely used delivery platform for cancer vaccines. Both RNA and DNA viruses including Newcastle disease virus (NDV), lentiviruses (LV), alphaviruses, flaviviruses, measles viruses (MV), rhabdoviruses, adenoviruses (Ad), adeno-associated viruses (AAV), herpes simplex viruses (HSV), poxviruses, and picornaviruses have been explored. These vectors can be designed to deliver oncolytic viruses, and the expression of immunostimulatory genes and TAAs has been investigated [89].

The distinct characteristics of each virus present unique advantages and challenges that can significantly influence its effectiveness in a specific therapeutic context. The production of host-induced neutralizing antibodies is among the limitations that restrict their future therapeutic utilization [88].

For efficient MHC class I antigen presentation, the antigen must be synthesized intracellularly to induce cytotoxic T-cell responses. Recombinant viral vectors such as adenovirus, parvovirus, vaccinia virus, lentivirus, and adeno-associated virus are genetically engineered to encode and deliver neoantigens. By effectively transducing cells, including professional APCs, these vectors express high amounts of the transgene and ultimately induce a robust immunological response from CD4^+^ and CD8^+^ T cells [90].

Adenoviruses are double-stranded, non-enveloped viruses with a 36 kbp genome that can accommodate 7.5 kbp of cDNA. The risk of insertional mutagenesis associated this vector is reduced since genome replication occurs in the nucleus but the vector remains extrachromosomal. By deleting the E1 region, the adenoviral vectors commonly used for immunization become replication-incompetent, reducing toxicity while still allowing humoral and cellular responses to transgenes. Low-level pathogenicity, genetic safety, strong immunogenicity, a lack of host genome integration, effective infection of various cell types, high transgene incorporation capacity, and relative ease of vector construction and production under Good Manufacturing Practice (GMP) are all characteristics of adenoviruses [91,92,93]. Viral-based cancer vaccines have shown promising effectiveness in preclinical and clinical studies.

### 2.6. Bacteria-Based Cancer Vaccine

The various applications of bacteria-based vaccine vectors include targeting immunosuppressive compounds, delivering cytokines, and vaccinating against changeable TAAs. Cancer vaccines are more effective in older patients when tumor-killing chemicals are delivered to the TME via bacterial vectors and eliminated or converted utilizing chemotherapeutics or attenuated Listeria [94]. It is well-known that bacteria, carcinogens, and areas that promote tumor growth are related. However, it has also been acknowledged that some bacterial species, including Salmonella, Escherichia coli, Lactococcus, Clostridium, Shigella, Bifidobacteria, Listeria, Vibrio, and Shigella, possess exceptional qualities that enable them to target and destroy cancer cells. These bacteria have shown promising potential for cancer therapy due to their ability to migrate toward hypoxic regions, stimulate immune responses, exhibit chemotaxis, and exert tumor-killing effects [95].

Moreover, previous studies have shown that photosynthetic bacteria, such as Salmonella sp. and E. coli, can effectively migrate to the hypoxic regions of tumor cells when administered intravenously [96]. The ability of bacterial species to transfer various therapeutic substances, including genes, proteins, DNA, and small molecules, to malignant cells has been well studied [97].

Live, attenuated, or genetically modified microbes, bacterial toxins as immunotoxins or conjugated to tumor antigens, bacteria-based cancer immunotherapy, bacterial vectors for gene-directed enzyme prodrugs, and the indisputable role of probiotics in treatment are some of the ways that bacteria are being used to treat cancer. Because bacterial therapy may lyse tumor cells and release therapeutic molecules, it has demonstrated higher potential for cancer treatment. The drawbacks of bacteriotherapy for cancer include the potential cytotoxicity of bacteria to healthy tissues, their inability to lyse malignant cells completely, and the potential for genome alterations [95].

The ability of bacteria to specifically target cancer cells, undergo genetic alterations to eliminate virulence characteristics, and grow within the body, thereby eliminating the need for recurrent injections, is a promising prospect for cancer therapy. One novel method of treating cancer that takes advantage of these traits is bacteriotherapy. Promising prospects for cancer therapy include bacteria’s capacity to multiply within the body, which eliminates the need for repeated injections; their ability to specifically target cancer cells; and their capacity to undergo genetic changes to eliminate virulence factors [98].

## 3. Tumor Microenvironment: Cellular Composition

Cancer cells interact dynamically with their surrounding microenvironment, composed of stromal cells (the cellular component) and extracellular matrix (ECM) components (the non-cellular part). These interactions promote tumor heterogeneity, clonal evolution, and multidrug resistance, ultimately leading to cancer progression and metastasis [99]. Manipulation of non-malignant cells by tumor cells, along with reciprocal cell–cell and cell–ECM interactions, causes stromal cells to lose their normal function and acquire tumor-promoting properties that facilitate cancer growth and dissemination. Understanding fundamental cellular and molecular processes underlying these interactions can help design safe and effective treatment strategies to combat cancer, as well as provide a novel way to target tumor-stroma communication [100].

The tumor microenvironment (TME) is composed of tumor cells, endothelial cells, tumor stromal cells (including stromal fibroblasts), immune cells (such as lymphocytes, macrophages, and microglia), and non-cellular ECM constituents (such as collagen, fibronectin, hyaluronan, and laminin) [100,101]. Tumor cells form the core of the TME and regulate the activity of both cellular and non-cellular ECM components via intricate signaling networks. This crosstalk contributes to multidrug resistance (MDR), poor therapeutic responses, and tumor development and maintenance. It is well known that non-malignant cells in TME encourage carcinogenesis throughout tumorigenesis [100].

The intricate cellular composition of the TME also affects the efficacy of cancer vaccines. Immune cells of TME, including T cells, B cells, and natural killer cells (NK), play several roles in regulating anti-tumor immune response. T cells display functional flexibility, allowing them to differentiate into cytotoxic effector T cells or into regulatory T cells (Tregs) that suppress immune responses. Their activities are modulated by immune checkpoint receptors, such as CTLA-4 and PD-1. Blocking these immune checkpoint receptors has become an effective strategy in cancer immunotherapy [11].

Similarly, B cells play numerous roles within the TME. They can produce antibodies, regulate the processing and presentation of antigens, and exhibit both pro- and anti-tumorigenic effects [102]. Regulatory B cells (Bregs) suppress antitumor immune responses by secreting IL-10 and IL-35 [103]. The dynamic interactions between TME immune cells and tumor cells strongly influence vaccine efficacy. TME components, including dendritic cells (DCs), neutrophils, tumor-associated macrophages (TAMs), and cancer-associated fibroblasts (CAFs), further contribute differently to tumor growth and immune regulation. However, targeting specific TME components, or pathways like fibroblast activation pathways or immune checkpoint signaling pathways, offers promising opportunities to overcome these barriers and enhance cancer vaccine-based immunotherapy [11].

In the liver, immunocompetent cells such as Kupffer cells (KCs), liver sinusoidal endothelial cells (LSECs), hepatic stellate cells (HSCs), pit cells, and lymphocytes, like NK T cells, γδ T cells, and DCs maintain immune tolerance due to constant exposure to gut-derived antigens [104]. However, chronic liver injury caused by hepatitis virus, parasitic infections, drugs, alcohol, and metabolic dysfunction disrupts this balance, leading to persistent inflammation, progressive fibrosis, and hepatocarcinogenesis [105].

During chronic inflammation, hepatic satellite cells (HSCs) transform into proliferative, contractile myofibroblasts that deposit large amounts of ECM proteins, such as collagens, fibronectin, and laminins [106]. This remodeled extracellular matrix (ECM) promotes tumor growth, invasion and metastasis, and profoundly alters the interaction among other TME components of HCC [107]. This resulting dense and fibrotic stroma increases tissue stiffness, distorts sinusoidal architecture, restricts vascular perfusion, and physically impedes the infiltration of cytotoxic CD8^+^ T cells and NK cells. It also creates hypoxic niches and chemokine gradients that further activate stromal cells and impair antitumor immunity [108].

The accumulation of various regulatory cell populations reinforces the immunosuppressive landscape of HCC. Within the TME, myeloid-derived suppressor cells (MDSCs), TAMs, tumor-associated neutrophils (TANs), Tregs, Bregs, and mature regulatory dendritic cells (mregDCs), collectively suppress cytotoxic lymphocytes through PD-1/PD-L1 and CTLA-4 engagement and by secreting IL-10, TGF-β, and VEGF. Tumor–derived exosomes and soluble mediators such as osteopontin (OPN) and IL-6 reprogram infiltrating neutrophils and macrophages into pro-tumoral phenotypes. Innate lymphoid cells (ILCs) further disrupt the immune balance. ILC2s and ILC3s secrete IL-13 and IL-17 to stimulate angiogenesis, fibrosis, and recruitment of suppressor cells, whereas NK cells and ILC1s remain cytotoxic. Together, these mechanisms impair antigen presentation, induce CD8^+^ T-cell exhaustion, and diminish NK cell activity, enabling HCC to escape immune surveillance [109].

The majority of mesenchymal cells in the HCC tumor microenvironment are cancer-associated fibroblasts (CAFs). These cells secrete diverse growth factors and cytokines that reinforce an immunosuppressive profibrotic environment and promote tumor growth, metastasis, and treatment resistance [107]. Through the IL6-STAT3 pathway, CAFs stimulate PDL1+ neutrophils, which promote immune suppression in HCC [110]. In addition, CAFs also remodel the ECM and directly interact with endothelial and immune cells, facilitating tumor angiogenesis while limiting cytotoxic T cell infiltration. This integrated immunosuppressive and fibrotic network not only fosters tumor progression but also contributes to resistance against conventional therapies and emerging immunotherapies. Thus, targeting CAFs or their downstream signaling pathways, such as IL-6 and STAT3, represents a promising strategy to overcome immune evasion and improve therapeutic outcomes in HCC [111].

## 4. Cancer Vaccine Resistance: Mechanisms

Many factors, including individual genetic variances, particularly in tumoral somatic cells, can contribute to cancer cells’ resistance to anticancer drugs. Drug resistance may also be acquired through various mechanisms, including multidrug resistance, apoptotic suppression, altered drug metabolism, epigenetic genes, enhanced DNA repair, and gene amplification [112].

Cancer vaccine resistance arises from a complex interplay of intrinsic and extrinsic mechanisms within the tumor and TME, encompassing several cellular and non-cellular components. Tumor cells can resist cancer vaccine-induced immunity through intrinsic mechanisms, such as cytokine/chemokine remodeling, antigen presentation defects, signaling pathway dysregulation, and immune editing, which together impair T-cell activation and recognition. Extrinsic resistance arises from stromal myeloid remodeling, checkpoint upregulation, and cytokine-driven suppression, which impair vaccine-induced immunity. Gaining a deeper understanding of these intrinsic and extrinsic mechanisms is essential for improving cancer vaccine efficacy [6,113].

TME is a major driver of drug resistance, relapse, and cancer incurability. Through interactions among tumor cells, stromal components, ECM, soluble mediators such as VEGF, IL-6, and GM-CSF, the TME provides survival signals that promote environment-mediated drug resistance (EM-DR). This includes both cell adhesion-mediated (CAM-DR) and soluble factor-mediated (SM-DR) mechanisms [112,114]. Drugs can change their molecular properties and become active when they interact with various proteins (in vivo). Cancer cells acquire resistance by changing drug uptake, metabolism, or signaling pathways, thereby reducing the overall effectiveness of the medication [115].

The majority of membrane transporters are solute carrier SLC transporters, which move minerals, vitamins, and other substances. These transporter proteins further influence resistance. There are two primary methods to decrease medication absorption, including lowering the number of transporters and/or the propensity for drug binding. Certain chemicals enter cells via specific transporters [116]. These transporters are inhibited by mutations, which lower drug absorption [112].

The most crucial elements in determining the quantity of the agent within and outside of cells are enzymes. Phase I processes, which include oxidation, reduction, and hydrolysis, and phase II reactions, which include consumption and conversion, are crucial in defending healthy cells from harmful substances. By decreasing prodrug activation (which lowers the activity of certain enzymes) and boosting drug inactivation (which raises the activity of other enzymes), these responses reduce drug resistance in cancer cells [117].

Overall, understanding the multifaceted mechanisms underlying drug and vaccine resistance, including genetic and molecular interactions within TME, is crucial for developing more effective, durable, and personalized cancer therapies.

## 5. Key Components and Factors Affecting the Efficiency of Cancer Vaccines

To effectively combat cancer, cancer therapeutic vaccines boost the formation of the body’s defenses through a complicated mechanism of action. It is a multifaceted procedure with multiple parts.

### 5.1. Selection of Target Antigens and Neoantigen-Based Cancer Immunotherapy

The most significant step in developing a cancer vaccine is selecting the antigens. A higher mutation burden enables cancer to develop immunological escape and drug resistance, but it also improves clinical outcomes with ICIs and strengthens specific T-cell responses. As a result, choosing and designing targeted antigens optimally is essential. Ideal antigens are those that are safe, highly immunogenic, tumor-specific, and applicable across a wide range of patients [6].

The antigen should ideally be present on all cancer cells, expressed exclusively by cancer cells, absent from normal tissues, essential for the survival of cancer cells, and highly immunogenic. Most of therapeutic cancer vaccines are built around one or a small number of cancer antigens [118].

Neoantigens arise from single-nucleotide variations, insertions and deletions, frameshifts, gene fusions, and human endogenous retroelements. Atypical proteasome processes, post-translational modifications (such as phosphorylation, glycosylation, and methylation), alternative translation involving noncanonical open reading frames (ORF), long non-coding RNA, and altered start codons, and aberrant transcriptions (such as splicing events, polyadenylation, and RNA editing) can also generate them [119,120]. Neoantigen identification process usually involves three steps: filtering and prioritizing candidate neoantigens, confirming their immunogenicity, and predicting neoantigens using human leukocyte antigen (HLA) typing [121].

Advances in next-generation sequencing and computational biology have enabled more affordable, widely available technologies, driven by developments in bioinformatics and high-throughput sequencing. By combining whole exome sequencing, RNA sequencing, and mass spectrometry data from tumor and matched normal tissues, potential tumor-specific mutations can be fully evaluated using a computational approach [120,122,123]. In addition to validating the expression of mutant genes, RNA-sequencing provides a wealth of biological information at the transcriptional level, including alternative splicing and changes in gene copy number [124].

The identification of noncanonical antigens is enabled by mass spectrometry, which directly detects aberrant peptides bound to HLA molecules [125]. Following the initial prediction, the list of potential neoantigens is filtered. A prioritized list of possibilities is produced by taking into account variables including expression level, dissimilarity to self-protein, mutation clonality, presentation efficacy, HLA binding affinity, and the stability of the peptide–HLA complex [126,127].

Neoantigen displayed on tumor cells can be specifically recognized by TCRs in the context of MHC molecules [128]. Neoantigens are powerful targets for the immune system to recognize and eliminate malignant cells. Because they lack central tolerance, neoantigens are more immunogenic than TAAs. Additionally, neoantigens represent a significant advancement in cancer immunotherapy, opening the possibility of customized and efficient tumor treatments [129].

Neoantigen-based vaccine development, production, and application could revolutionize cancer therapy modalities and improve patient outcomes [130]. Tumor neoantigens arise from various somatic alterations that generate novel peptides recognizable by the immune system. Viral proteins in virus-induced cancers, splicing site mutations, gene fusions, non-synonymous mutations (SNVs/INDELs), aberrant transcription or translation events (non-coding RNA translation, intron retention, alternative ORFs), and structural variants are major sources of tumor-specific immune targets for cancer immunotherapy [131].

Many nonsynonymous genetic changes, such as single-nucleotide variations (SNVs), insertions and deletions (indel), gene fusions, frameshift mutations, and structural variants (SVs), may be the source of tumor neoantigens for the majority of human malignancies that do not have a viral etiology. For malignancies linked to viruses, such as cervical or oropharyngeal cancer caused by the human papillomavirus (HPV), Merkel cell carcinoma (MCC) caused by the Merkel cell polyomavirus (MCPyV), and head and neck cancers caused by the Epstein–Barr virus (EBV), any epitopes come from open reading frame [132].

Neoantigen-based immunotherapy is a promising approach in personalized cancer treatment. This approach focuses on tumor-specific genetic changes to generate unique immunogenic peptides that can be distinguished from normal self-proteins. These neoepitopes, which arise from somatic mutations, are processed and presented by HLA molecules on the cell surface, where they can activate potent T cell mediated immune responses against the tumor [133].

Neoantigen discovery has been significantly accelerated by next-generation sequencing and advanced computer modeling. This has enabled high throughput forecasting of candidate epitopes using machine learning methods like NetMHCpan, NetMHCIIpan, MHCflurry, ConvMHC, PLAtEAU, and NetCTLpan. Nevertheless, only a small percentage of anticipated peptides elicit immunological responses, despite tremendous advancements. The complex mechanisms underlying antigen presentation and limited datasets are the main causes of these shortcomings. Recent developments in HLA peptidomics and deep learning tools like EDGE have significantly improved dataset quality and prediction accuracy, helping overcome these restrictions. In addition, integrative algorithms like MuPeXI, EpitopeHunter, and Neo-pepsee now combine genomic, transcriptomic, and structural data to provide a more precise evaluation of immunogenicity. When combined, these developments are boosting the precision and effectiveness of neoantigen-based cancer immunotherapies by advancing the precise identification of clinically relevant neoepitopes [128].

Novel immunotherapeutic approaches targeting tumor neoantigens, such as neo-antigen-based vaccines, neoCART therapies, and neoantigen-specific antibodies, are showing great promise in inducing durability antitumor immunity and preventing recurrence or metastasis. Personalized neoantigen identification and vaccine production have been accelerated by developments in high-throughput sequencing and AI-driven prediction algorithms. However, there are still problems with costs, accuracy, and production time optimization. Ongoing integration of biotechnology, immunology, bioinformatics, and AI continues to enhance neoantigen discovery and the development of next-generation immunotherapies [131].

New technologies have enabled the development of more potent cancer vaccines targeting neoantigens, which trigger immune responses that destroy cancer cells [134]. There are two primary categories of neoantigen-based cancer vaccines: off-the-shelf and customized [135,136]. A wider range of cancer patients can benefit from off-the-shelf vaccinations because they are designed to target shared neoantigens, which are expected to be frequently expressed and elicit strong anti-tumor immune responses [137]. On the other hand, customized vaccines are developed using the distinct neoantigens found in each patient’s tumor. Personalized vaccinations induce powerful, and targeted anti-tumor immune responses and immunological memory, precisely eradicating cancer cells without endangering healthy tissues, because of their high anti-tumor specificity and low central immune tolerance [131].

A multidisciplinary strategy, including genomic sequencing, bioinformatics, vaccine design and manufacture, preclinical and clinical testing of vaccine performance, and regulatory approval for clinical use, is necessary for the effective design of a neoantigen cancer vaccine. The development of a safe and efficient neoantigen cancer vaccine has enormous potential to improve the results of cancer treatment, despite the difficulties associated with this approach [138].

Developing a neoantigen-based cancer vaccine involves a series of well-defined steps. First, tumor-specific neoantigens are identified and characterized using advanced genomic sequencing and bioinformatics to detect unique patient-specific mutations. Next, the vaccine is designed and formulated by selecting the most immunogenic targets and enhancing their stability and potency with optimized delivery systems. These selected neoantigens are then used to manufacture vaccines under good manufacturing practice conditions for clinical use. Subsequently, preclinical evaluations are conducted to assess safety, efficacy, and immunogenicity using both in vitro assays and animal tumor models. Finally, clinical trials are conducted to validate the vaccine’s safety and therapeutic effectiveness in human patients [131].

### 5.2. Activation of Immune Response

Cancer vaccines induce antigen-specific T-cell responses to target tumor cells [139]. Immune checkpoint blockade and adoptive T cell therapies similarly induce T-cell activation [140]. Activated T cells differentiate into cytotoxic, helper, and memory subsets, while B cells mature into plasma and memory cells, producing tumor-specific antibodies that mediate CDC and ADCC. Cytotoxic T cells kill tumor cells via perforin/granzyme or FAS ligand, and tumor-infiltrating APCs further enhance immune response [6,141].

### 5.3. Role of Adjuvants

Adjuvants are vital components of vaccines that increase the immune responses. Adjuvants are classified into immune-stimulants and delivery systems. Immuno-stimulants act as danger signals that activate and mature APCs through toll-like receptors and other pattern-recognition receptors. Thereby, they improve adaptive immunity. Delivery systems serve as carriers, which increase antigen bioavailability and facilitate targeted delivery to lymph nodes or APCs for efficient immune activation [142]. Although overall survival has increased significantly as a result of cancer treatment, immune system evasion remains a problem for patients with established disease burdens. Approaches that use combination adjuvants with diverse mechanisms may be helpful, because it can be challenging to stimulate an immune response against cancer [143].

Adjuvants are essential for increasing the efficacy of cancer vaccines and can be roughly divided into three groups, including immunomodulatory compounds, adjuvant-containing delivery vehicles, and mixtures of the two [144]. They improve immune responses by mimicking pathogen-associated molecular patterns, which trigger the release of damage-associated molecular patterns (DAMPs), enhance APC activation, increase antigen bioavailability, and support efficient antigen transport [145]. Delivery systems use both physical and chemical methods to improve targeting, increase bioavailability, and shield antigens from deterioration. Electroporation, gene guns, and microneedles are examples of physical delivery technologies that primarily serve as mechanical devices for the effective delivery of antigens [6,146,147].

Water-in-oil emulsions, lipid nanoparticles (LNPs), polymeric particles, and nanomaterials are examples of chemical delivery systems frequently employed. Some of them possess inherent immunostimulatory properties, making them make them suitable as adjuvants [142].

### 5.4. Administration Routes of Cancer Vaccines

Various routes of administration, such as intramuscular, intradermal, subcutaneous, intravenous, oral, intra-tumoral, and mucosal, have distinct effects on the immunological response, overall safety, and effectiveness of cancer vaccines [148]. Compared to the intramuscular approach, the subcutaneous route was shown to improve nanoparticle distribution, facilitate lymph node drainage and induce more neoantigen-specific T cells [149]. Both intravenous and subcutaneous vaccination elicited particular tumor-infiltrating T lymphocytes (TILs) in tumor-bearing animals, but only the intravenous approach led to measurable tumor regression and a reduction in regulatory monocytes [150].

Mucosal delivery methods such as sublingual, intranasal, and atomization have the potential to induce mucosal immunity. Because of their direct cytotoxicity capabilities and strategic tissue localization, tissue-resident memory T cells (TRMs), which are preferentially produced by mucosal vaccination, are appealing biomarkers linked with better survival rates [151,152,153].

## 6. Engineering Strategies to Enhance Tumor Cell-Derived Vaccine

Altered vaccine engineering techniques can improve therapeutic efficacy, targeting precision and delivery efficiency. The therapeutic efficacy of vaccines derived from tumor cells can be flexibly enhanced by various engineering modifications. These engineering techniques can significantly improve immunogenicity of tumor vaccines. Vaccines against cancer use a variety of strategies to boost immunity and produce a potent antitumor reaction.

### 6.1. Genetic Engineering

This technique may be used to introduce exogenous genes that modify cellular phenotypes and increase immunogenicity. Techniques include the knockout, insertion, or replacement of nucleic acid sequences via gene editing, viral/non-viral vectors, or physical methods such as electroporation and microinjection. Viral vectors (e.g., adenovirus, lentivirus) provide high efficiency but risk insertional mutations, whereas non-viral carriers (lipid- or polymer-based nanoparticles) are safer but less effective. Physical methods for delivering nucleic acids depend on altering cell membrane permeability, though their efficiency is limited. CRISPR/Cas9 and mRNA delivery systems offer greater flexibility, allow rapid expression, and enable controlled levels of target molecules. For example, lentivirus-mediated transfection of α-lactalbumin mRNA into tumor cell-derived exosomes, thereby improving their targeting ability. Overall, genetic engineering allows for selective, significant and controlled levels of target molecules [70].

### 6.2. Surface Engineering

Surface engineering modifies the membrane of tumor cells or exosomes to improve delivery and target immune activation. Chemical approaches, such as amination and click chemistry, couple functional groups (amines, carboxyls, azides, alkynes) with targeting ligands, enabling tumor-specific delivery. However, these reactions may damage biological components and are time-intensive. Physical methods, including electrostatic and hydrophobic interactions, offer milder and faster alternatives by attaching molecules such as mannose, antibodies, or anchoring ligands (e.g., DSPE-PEG) to membranes. Protein anchoring via glycosylphophatidylinositol (GPI) or the avidin–biotin interaction also enhances vaccine functionality. While chemical coupling offers stability, it is limited by ligand choice and potential toxicity; physical adsorption is simpler but less stable. Overall, genetic engineering allows for selective, significant and controlled levels of target molecules to boost vaccine efficacy [70].

### 6.3. Internal Cargo Loading

Internal cargo loading can involve adding tumor cell-derived vaccines to the interior of other carriers (non-tumor-origin biomaterials such as hydrogels and organic/inorganic nanoparticles) or encapsulating exogenous materials (such as drug molecules, photosensitizers, and inorganic/organic biomaterials) within tumor cell-derived carriers. The internal capacity of extracellular and cell membrane vesicles makes them suitable for use in this technique. Since RNA vaccines cannot be employed as a vehicle on their own, the latter approach is better suited for delivering cell lysates. When exogenous materials are added to tumor cell-derived vesicles, they can enhance their immunogenicity and photothermal properties, thereby improving their therapeutic efficacy against cancer cells [104]. Depending on cell type, diffusion, endocytosis, and electroporation can be used to load internal cargo into tumor cells [154].

## 7. Delivery Methods

To obtain favorable clinical outcomes, cancer vaccines should combine the best tumor antigens with effective immunotherapeutic drugs and/or delivery systems. Multiple vaccination routes and physical/chemical delivery techniques have been explored to enhance immune responses against TSAs, with preclinical studies highlighting microparticle-based targeting of APCs and physical methods that increase antigen expression. Ongoing advances in vaccine administration technologies will improve clinical trial success and efficiency [155]. Figure 3 presents an overview of the different delivery methods. Table 5 provides an overview of delivery approaches for cancer vaccines, including their underlying principles, methodologies, advantages, and limitations.

**Figure 3 cimb-47-01056-f003:**
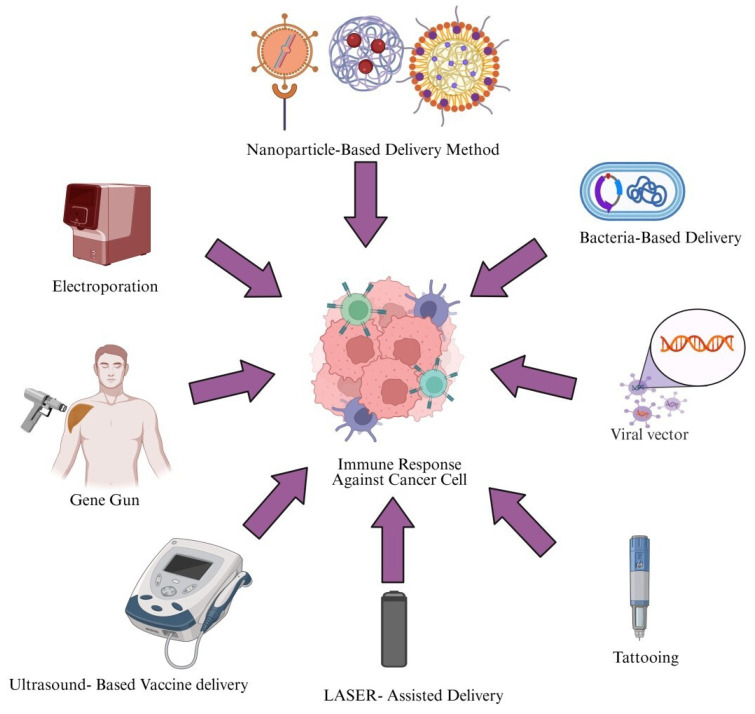
Delivery approaches for cancer vaccines. This figure illustrates various vaccine delivery methods used to induce an immune response against cancer cells. Created in https://BioRender.com (accessed on 2 October 2025).

**Table 5 cimb-47-01056-t005:** Overview of delivery approaches for cancer vaccines, including their underlying principles, methodologies, advantages, and limitations.

Method	Principle	Methodology	Advantages	Limitations	References
Electroporation	Introduce nucleic acids into cells either in vitro or in vivo by applying electric pulse to induce temporary and reversible permeabilization of cell membrane	DNA plasmid is injected intramuscularly or intradermally, short electric pulses applied via electrodes	High efficiency	Painful, possible damage to the skin and muscle tissue at the injection site	[156]
Gene-gun delivery methods	DNA-coated microparticles penetrate cells via high pressure gas burst	Plasmid DNA precipitated on gold/tungsten particles loaded in cartridges and shot into skin or tissue using helium pressure.	Low cytotoxicity, less tissue damage	Less efficiency	[157]
Ultrasound (Sonoporation)	Cavitaton induced by ultrasound lead to pore formation and ultimately allows nucleic acid entry	DNA/NP suspension applied to tissue/cells, ultrasound generates microbubbles facilitating uptake	Non-invasive, enhance APC activation	Optimization needed	[158,159]
LASER assisted Delivery	Low energy LASER waves cause micropore formations, allowing nucleic acid entry	Target tissue irradiated with LASER pulse before and after DNA application and DNA becomes diffused into permeabilized cells	High efficiency, localized targeting	Tissue heating	[160]
Tattooing	Rapid puncturing by tattoo needles delivers DNA intradermally into APC-rich skin	DNA solution applied, tattoo machine punctures skin, delivering DNA directly to dermis	Strong immune response, fast delivery	Pain, cosmetic concern	[161]
Viral vector mediated method	Recombinant virus deliver antigen encoding genes into host cells	Genes inserted into adenovirus or lentivirus are injected to infect APCs or tumor cells	High immunogenicity, persistent response	Safety risk, might be pre-existing immunity present	[162]
Bacterial vectors mediated delivery	Attenuated bacteria invade APCs and deliver antigens genes/proteins	Listeria/salmonella engineered with tumor antigens administrated systemically or orally and stimulate innate and adaptive immunity	Tumor targeting, strong activation	Safety and toxicity issues	[163]
Nanoparticle based delivery method	Nanocarriers encapsulate and protect DMA/RNA/Protein enabling controlled release	Lipid/polymer nanoparticles prepared by solvent methods encapsulate antigens and are injected for APC uptake	Stability, controlled release	Complex manufacturing	[164]

### 7.1. Electroporation

Electroporation transiently increases cell membrane permeability through brief electrical pulses, facilitating intracellular plasmid DNA uptake. By accelerating 1000-fold DNA uptake in some studies, electroporation also induces local inflammation, promotes cytokine release, and recruits APCs, thereby acting as an intrinsic adjuvant [165].

EP significantly enhances both humoral and cellular immune responses across multiple animal models and supports clinical translation. Comparative studies have shown that EP outperforms ultrasound in transfecting naked plasmid DNA [155].

### 7.2. Gene Gun

One non-viral way to introduce genes into cells is to use a gene gun (particle-mediated) to deliver a DNA vaccine. This application enables the intradermal delivery of gold particles coated with DNA. Compared to intramuscular needle injections or the Biojector system, this method produces a greater quantity of CTLs [166]. This approach lowers the amount of DNA needed 100–1000-fold [167] and has advanced to phase 1 and 2 clinical trials in cancers, including cervical cancer and head and neck squamous cell carcinoma have based on encouraging preclinical evidence [168].

### 7.3. Ultrasound-Based Method

Ultrasound therapy facilitates DNA vaccine delivery by temporarily disrupting cell membranes via mechanical and cavitation effects. Preclinical studies report immune responses up to 10-fold higher than those achieved with subcutaneous injection, alongside enhanced activation of Langerhans cells [166,169,170]. Ultrasound exposure can increase tumor immunogenicity, aiding antigen penetration and transforming immunologically “cold” tumors into “hot” tumors, thereby increasing the immunotherapy effect [171].

### 7.4. LASER-Assisted Delivery Method

This new approach increases the intradermal plasmid transfection and induces both humoral immunity and an antigen-specific CD4^+^ and CD8^+^ T-cell response. Although limited studies are available, several reports highlight its high potential for therapeutic HPV DNA vaccination [172]. The clinical translation of this technique into intradermal or transdermal vaccination might be accelerated by ongoing expansion of intradermal vaccination platforms and the medical LASER field, with emerging data also suggesting that LASER adjuvants can enhance the effectiveness of immunotherapy in allergic diseases [173].

### 7.5. Tattooing

DNA tattooing introduces DNA into skin cells in a short time through accelerated immune activation. Compared with intradermal injection or gene gun delivery, tattooing yields higher gene expression patterns [148]. In a mouse model, tattooing 20 μg of DNA achieved substantial gene expression levels compared with intra-muscular injection of 100 μg of DNA [174].

### 7.6. Biological Methods

#### 7.6.1. Viral Systems

Viruses are widely used as a DNA vaccine delivery system due to their natural infectivity and efficient gene transfer into the cell genome [166]. Many viral vectors have been developed with excellent efficacy, minimum toxicity, and immunogenicity. Platforms such as adenoviruses, poxviruses, and herpes simplex viruses enable high antigen expression and strong immunogenicity, with demonstrated clinical efficacy in solid tumors including melanoma and glioblastoma [166,175,176,177,178,179].

Clinical studies have shown promising outcomes, including phase I data on oncolytic HSV VG161 in advanced HCC [180]. A Newcastle Disease Virus (NDV)-based vaccine engineered to induce αGal antigen expression enhanced antitumor immunity by mimicking xenograft rejection. Clinically, it showed notable efficacy in advanced, treatment-refractory solid tumors, achieving a 90% disease control rate [181].

#### 7.6.2. Bacterial Delivery System

Bacteria-based delivery systems have gained attention for the detection and treatment of cancer and other diseases due to their biocompatibility, tumor-targeting ability, immuno-stimulatory properties, and motility [181]. Although clinical applications is limited by incomplete understanding tumor-targeting mechanisms, limited antitumor efficiency, and biosafety concerns [182], live bacterial vehicles remain promising for cancer immunotherapy due to their ability to selectively colonize tumors, enable precise drug delivery, and immune activation [5]. Facultative and obligate anaerobic bacteria, including Clostridium, Bifidobacterium, Salmonella typhimurium, Listeria, and Escherichia coli can be genetically engineered or surface-modified to deliver immune factors, drug molecules, and nucleic acids, within TME [183], thereby inducing tumor cell autophagy and apoptosis, while stimulating host immunity [184,185]. Their active motility, particularly in flagellated species, helps overcome the limited tissue penetration associated with a conventional passive delivery system [183].

### 7.7. Nanoparticle-Based Delivery of Cancer Vaccines

Exosomes and nanoparticles (NPs) are widely utilized for targeted drug delivery due to their nanoscale size, biocompatibility, and ability to enable targeted distribution, controlled release, and enhanced antigen stability, thereby improving therapeutic efficacy and immune activation [11,186,187]. Lipid, polymeric, virus-like, and inorganic NPs, as well as exosomes, can be engineered to deliver antigens, adjuvants, nucleic acids, including mRNA, directly to immune or tumor cells, reducing systemic toxicity and strengthening antitumor immune response [188,189,190,191,192].

Lipid NPs are particularly suited for mRNA cancer vaccines, supporting antigen customization, and clinical translation, while liposomes and polymeric carriers, including poly (lactic-co-glycolic acid) (PLGA), polyethylene glycol (PEG), polycaprolactone, chitosan, and dextran enhance immunogenicity through tunable physiochemical properties [190,193,194,195,196]. Nanotechnology-enabled in situ vaccination further exploits the TME to induce immunogenic cell death and convert immunologically cold tumors into hot tumors [197]. In addition, inorganic nanocarriers such as gold nanoparticles have demonstrated efficient lymph node targeting and robust CD8^+^ T cell stimulation, highlighting the broad translational potential of NP-based cancer vaccine platforms [198,199].

## 8. Clinical Applications of Cancer Vaccines in Hepatocellular Carcinoma

HCC is the third leading cause of cancer-related mortality worldwide, and it is a major global health burden. Approximately 90% of primary liver malignancies are HCC. However, intrahepatic cholangiocarcinoma and other less common forms rank second and third in this sequence. Despite advances in medical care, the 5-year survival rate remains low at around 5–6%, mainly due to limited therapeutic options and late diagnosis. Numerous factors, such as genetic variations, gut microbial dysbiosis, aging, obesity, gender, and alcohol consumption, all contribute to liver cirrhosis and tumor development. Furthermore, chronic viral infections such as hepatitis B virus (HBV) and hepatitis C virus (HCV) are well-known causes of liver carcinogenesis; in contrast, hepatitis D virus (HDV) speeds up the progression of severe liver disease in populations infected with hepatitis B virus (HBV) [3].

HCC develops within a chronic inflammatory setting, yet the liver maintains a markedly suppressive microenvironment. Various liver-resident cells, such as immune and stromal cells, sinusoidal endothelial cells (LSECs), Kupffer cells, and liver dendritic cells (DCs) establish intrahepatic tolerogenicity. However, hepatocytes can stimulate T-cell anergy or deletion, while myeloid-derived suppressor cells (MDSCs) and regulatory T cells (Tregs) further suppress the activity of cytotoxic T lymphocyte (CTLs) and natural killer (NK) cells [200].

HCC develops and progresses through interactions with the immune system and innate immune cells, such as Kupffer cells, TAMs, and NK cells, which are often dysregulated. This allows tumor cells to escape immune surveillance. At the same time, adaptive immune responses weaken due to the expansion of Tregs and the exhaustion of cytotoxic CD8^+^ T cells, reducing the body’s ability to target and destroy cancer cells. Key signaling pathways, such as TGF-β, IL-6/STAT3, and NF-ҝβ, drive a microenvironment that facilitates tumor growth, angiogenesis, and metastasis [201,202].

Tumors also exploit immune checkpoint molecules, such as PD-1/PD-L1 and CTLA-4, to suppress T-cell activity. This has led to the development of immune checkpoint inhibitors as targeted therapies. New strategies aim to present TSAs and activate DCs, thereby stimulating a stronger and more durable antitumor immune response [203]. Understanding these mechanisms provides a foundation for designing effective cancer vaccines and combinational immunotherapies that can overcome immune evasion and improve patient outcomes.

The inhibitory microenvironment in HCC limits effective immunity, but cancer vaccines may overcome this by boosting activating cytokines and reducing suppressive immune-cell activity. By enhancing pro-inflammatory cytokines and reducing the activity of immune regulatory cell subsets, cancer vaccines can mitigate these microenvironmental constraints. Long-lasting immunological memory may be another benefit of vaccination. When subsequent interactions with HCC cells occur, memory T cells generated after vaccination can provide prolonged monitoring and prompt responses, thereby limiting tumor recurrence and aiding long-term disease management [204].

By augmenting natural killer (NK) cells and cytotoxic T lymphocytes (CTLs), cancer vaccines enhance antitumor immunity [204,205]. In addition, they can also counter microenvironmental suppression by stimulating cytokine production, reducing suppressive immune cell populations, and generating long-lasting immunological memory. Memory T cells generated after vaccination can surveil and rapidly respond to recurring HCC cells, reducing recurrence and supporting long-term disease management [204].

Over the last 15 years, many TAAs, some of which trigger tumor-specific immune responses, have been discovered in HCC [206]. However, only a small number of these TAAs have been employed in clinical applications, as most of them are not specific to HCC. These include Wilms’ tumor 1 (WT-1) [207,208], telomerase reverse transcriptase (TERT) [209,210], alpha fetoprotein (AFP) [211], glypican 3 (GPC3) [212,213], MAGE-A, SSX-2, and NY-ESO-1 [214].

Interestingly, anti-TAA-specific T cells can be detected even in unimmunized patients, yet overall clinical outcomes have often been disappointing. The investigated TAAs are not only overexpressed in HCC but also seem to be immunogenic in patients, providing compelling justification for these clinical trials. Human telomerase reverse transcriptase (hTERT), alpha-fetoprotein (AFP), glypican-3 (GPC3), multidrug resistance-associated protein 3 (MRP-3), the cancer–testis antigens melanoma-associated antigen 1 (MAGE-1), and New York esophageal squamous cell carcinoma-1 (NY-ESO-1) are among the clinically tested TAAs that are overexpressed in HCC tumor cells. Additionally, unimmunized patients have anti-TAA-specific T cells, which were produced after vaccination. Despite this data, clinical outcomes fell short of expectations [215].

Encouraging immunologic responses have been observed in early clinical studies of alpha-fetoprotein (AFP)–targeted immunotherapy. A phase I trial demonstrated that AFP-specific genetic immunotherapy elicited detectable AFP-reactive cytotoxic T-cell responses in patients with hepatocellular carcinoma, confirming the feasibility and immunogenicity of this approach, although no objective tumor regression was reported [216]. Similarly, an early-phase clinical study of a glypican-3 (GPC3)–derived peptide vaccine in patients with advanced hepatocellular carcinoma demonstrated that the vaccine was safe and well tolerated, elicited GPC3-specific cytotoxic T-cell responses, and that stronger immune responses were associated with improved overall survival [217].

To enhance the therapeutic effectiveness of immunotherapy, combinatorial strategies such as adaptive cellular therapy, immune checkpoint suppression, cancer vaccines, and immune-activating cytokines are being explored [218]. Nearly all tumor types, including solid and blood malignancies at all stages, can be treated with a potent cancer vaccine platform targeting TAAs [219].

In early-stage cancer patients, vaccines may be used alone to induce targeted antitumor immunity and prevent recurrence. In the intermediate stage, cancer vaccines can be combined with other immunotherapeutic strategies to achieve a partial response or a complete response. When used in conjunction with other therapeutic approaches, cancer vaccinations help patients in their advanced stages live longer. The novel cancer vaccine platform aims to improve adaptive immunity in three ways. In the first approach, tumor antigens are modified to become soluble to enhance B cell activation and antibody production. In the second approach, antigens link to MHC-II pathway molecules such as LAMP-1, DCLAMP, and Ii signals, to improve antitumor CD4^+^ T-cell priming. In the third and last approach, antigens bind to MHC-I pathway molecules such as ubiquitin, p62, MITD, γ-tubulin, and the DD signal, to improve antitumor CD8^+^ T-cell priming [215]. These innovations are expected to increase antigen presentation efficiency and promote stronger, long-lasting antitumor immunity in HCC.

Here are several clinical/translational studies of vaccination approaches in HCC with reported outcomes (Table 6). These vaccines primarily target TAAs such as AFP and GPC3, leading to the induction of antigen-specific cytotoxic T lymphocytes (CTLs) and helper T cells. They expand AFP-specific T cells, enhance interferon ℽ production, and elicit both class I and II tumor antigen responses. Furthermore, vaccine-induced activation of CD4^+^ and CD8^+^ cells augments tumor-specific immunity while reducing regulatory T cell-mediated suppression. Collectively, these effects contribute to enhanced antigen presentation, improved immune surveillance, and potential survival benefits in HCC patients [220,221,222,223,224,225,226]. Figure 4 summarizes various immunological mechanisms of cancer vaccines in hepatocellular carcinoma. Clinical trials of cancer vaccines for hepatocellular carcinoma are summarized in Table 7.

**Figure 4 cimb-47-01056-f004:**
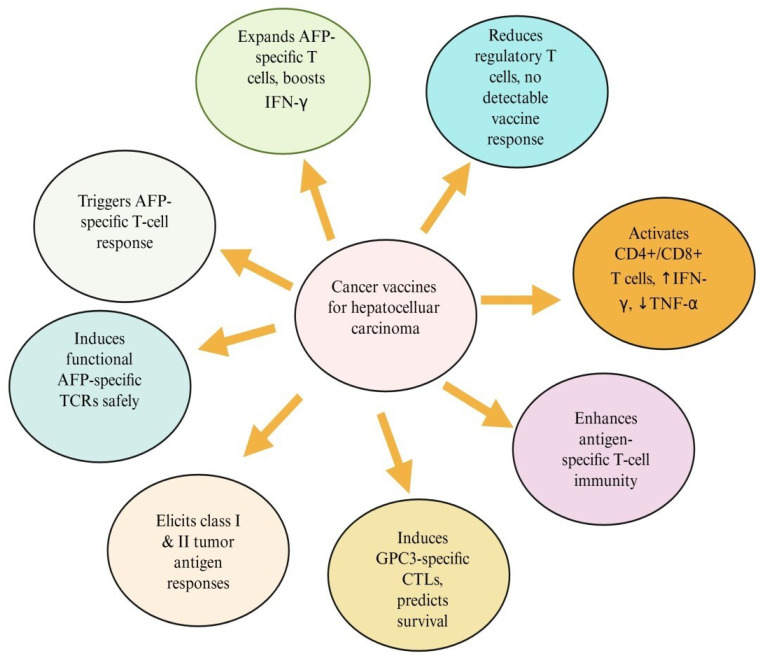
Immunological mechanisms of cancer vaccines in hepatocellular vaccines. Cancer vaccines targeting AFP and GPC3 peptide induce antigen-specific T-cell immunity, activate CD4^+^ and CD8^+^ responses, and modulate cytokine balance to enhance antitumor effects. Created at https://BioRender.com (accessed on 2 October 2025).

### 8.1. HCC-Specific Tumor Antigens and Their Molecular Characteristics

#### 8.1.1. Glypican-3

Glypican-3 (GPC3) is a TAA with comparatively low levels in normal tissues and specific expression in HCC [227]. Through multiple signaling pathways, including Wnt, IGF, YAP, and Hedgehog, GPC-3 participates in critical cellular processes such as proliferation, metastasis, apoptosis, and the epithelial–mesenchymal transition [228]. GPC3 is an oncofetal heparan sulfate (HS) glycoprotein that is bound to the cell membrane by a glycophosphatidylinositol (GPI) anchor. The GPC3 core protein is 70 kDa and contains 580 amino acids. Near the C-terminal region, two HS side chains are joined. Furin processes the single-chain GPC3 at the Arg358-Cys359 bond to produce the mature GPC3, which is made up of a 40 kDa N-terminal subunit and a 30 kDa C-terminal subunit connected by disulfide bonds [229].

GPC3 is a potential target for precision therapy and drug development in HCC due to its distinct expression pattern. Immunotherapies and GPC3-targeted therapies have advanced significantly in recent studies, especially for patients with advanced or treatment-resistant HCC. Numerous ongoing studies are investigating its therapeutic efficacy, despite some clinical trials yielding less-than-ideal results [227].

#### 8.1.2. Alpha-Fetoprotein

The liver and yolk sac produce the glycoprotein alpha-fetoprotein (AFP) during fetal development. Nonetheless, there is a strong association between the onset and progression of HCC in adults and serum AFP levels. There are 609 amino acid residues in AFP’s single-chain polypeptide. The N-terminal domain (1–210 amino acid residues; domain I), the central domain (211–402 amino acid residues; domain II), and the C-terminal domain (403–609; domain III) are the three structural domains formed by the arrangement of these amino acid residues. Disulfide bonds bind these three AFP domains together, creating a V-shaped structure [230].

AFP regulates cell proliferation and may be mediated by receptors, signal transduction, and gene expression. According to some studies, AFP is mainly found on the cell surface or in the cytoplasm, where it is internalized via receptor-mediated endocytosis. The cAMP-PKA pathway and the induction of Ca^2+^ influx are necessary for the promotion of AFP/AFPR tumor proliferation. Because AFP/AFPR causes Ca^2+^ influx, it increases intracellular Ca^2+^, which in turn raises intracellular CAMP, boosting protease activity. Tumor cell proliferation is driven through a mechanism that stimulates DNA synthesis [231].

Through multiple processes, including cell proliferation, angiogenesis, EMT, metabolic reprogramming, immune evasion, and drug resistance, AFP accelerates HCC development. By blocking PTEN activity and activating the PI3K/Akt pathway, AFP promotes the malignant behavior of HCC cells and induces the expression of proteins essential for cell metastasis, including keratin 19, matrix metalloproteinases 2/9, and CXCR4. Furthermore, AFP is linked to several microRNAs (miRNAs) that regulate EMT and affect important cellular functions such as invasion, migration, and metastasis [232].

#### 8.1.3. Human Telomerase Reverse Transcriptase (hTERT)

Human telomerase reverse transcriptase (hTERT) serves as a catalytic enzyme that promotes telomere elongation. It is found in numerous tumors, including HCC [233]. Several studies have demonstrated that TERT contributes to development of cancer through a variety of mechanisms, including regulation of T cell function, especially in virus-infected cells, in addition to its telomere-lengthening role. Due to liver damage and constant cell renewal, cirrhosis causes telomeres to become extremely short, which triggers the DNA damage response, cell senescence, and telomerase activation. Over 90% of HCC cases have TERT expression and telomere elongation, which have been linked to tumor aggressiveness and a poorer prognosis for patients. Furthermore, telomerase is essential for maintaining cell immortality and controlling the replicative program of some viruses. Virus-encoded factors regulate telomerase activity at several levels, including transcription, alternative splicing, post-transcription modifications, and subcellular localization [234].

Telomerase is a ribonucleoprotein polymerase that adds the telomere repeat TTAGGG to chromosome ends to keep them intact. The enzyme consists of an RNA component that serves as a template for the telomere repeats and a protein component with reverse transcriptase activity. Cellular senescence is influenced by telomerase expression, which is typically suppressed in postnatal somatic cells, leading to gradual telomere shortening [235].

#### 8.1.4. Cancer–Testis Antigens

A new class of TSAs known as cancer–testis (CT) antigens, including MAGE superfamily, SSX family, GAGE, BAGE, SCP-1, NY-ESO-1, CTp11, and HCA587, has recently been identified through the integration of molecular and immunological techniques [236]. They are categorized as testis-restricted, testis/brain-restricted, and testis-selective CTAs based on their expression profile in adult healthy tissues; the latter group exhibits additional expression in somatic tissues. Testis-restricted CTAs are thought to be perfect targets for cancer immunotherapy because they are not expressed in healthy adult tissues and may trigger antitumor immune responses [237]. The products of the CT antigen genes (MAGE-1, SSX-1, CTp11, and HCA587) are promising targets for antigen-specific immunotherapy of HCC due to their high expression rates and specificity. Polyvalent vaccinations for HCC are feasible because several CT antigens are coexpressed at high frequency [236]. The high frequency and specificity of CTAs in HCC suggest that their products could be novel, promising targets for HCC antigen-specific immunotherapy. When specific CTAs were detected in AFP-negative HCC, mRNAs were proposed as tumor markers to identify circulating HCC cells as an adjuvant diagnostic tool, as well as indicators of prognosis and recurrence [238].

#### 8.1.5. Other Emerging Antigens

The viability of tailored neoantigen-based vaccines in HCC has been supported by recent multi-omics and immunopeptidomics studies that have identified naturally presented neoepitopes in HCC, including those derived from driver-mutated genes such as TP53 and CTNNB1 [239]. Additionally, preclinical murine models show that neoantigen-based vaccination improves antitumor efficacy, especially when paired with immune checkpoint blockade [240]. Such mutation-derived antigens are likely to be patient-specific rather than universally shared, given the relatively low mutational load in HCC, highlighting the need for customized vaccine design [241].

## 9. Molecular Targets of HCC Treatment

Targeting key players in signaling pathways with molecular therapeutics has become possible through recent studies of the molecular signaling pathways in several cancers. Numerous molecular targets linked to the development and progression of HCC are reported. By regulating corresponding signaling pathways, several key cytokines, including TGF-β, PI3K, protein kinase B (Akt), VEGF, and NF-κB, can control the development, proliferation, invasion, metastasis, and apoptosis of HCC cells. Furthermore, numerous studies have demonstrated a tight relationship between the development of HCC and the primary renin-angiotensin system (RAS) components, including Ang II, ACE, ACE2, MasR, AT1R, and AT2R [242].

To develop novel and promising therapies with higher antitumor effects, recent progress in identifying molecular targets and developing immunotherapeutic approaches for HCC is reviewed. The molecular targets discussed include the intracellular signaling pathways of protein kinase B/mammalian target of rapamycin, RAS/RAF/mitogen-activated protein kinase, Wnt/β-catenin, glutamine synthetase, insulin-like growth factor, signal transducer and activator of transcription 3, nuclear factor-κB, telomerase reverse transcriptase, and c-MET. Immunological studies have focused mainly on target identification, T cells, natural killer cells, dendritic cells, natural killer T cells, and vaccine development [243].

Numerous potential molecular targets in HCC have been identified by growing understanding of the molecular pathways underlying tumor genesis and development. These targets include the Jak/Stat, PI-3K/Akt/mTOR, and Raf/MEK/ERK receptor tyrosine kinase-activated pathways. Several approaches have been developed and evaluated at different stages of clinical trials, including monoclonal antibodies and tyrosine kinase inhibitors including erlotinib, sunitinib, vandetanib, cediranib, brivanib, foretinib, and dovitinib [244].

## 10. Combinational Approaches to Enhance Cancer Vaccine Efficacy for HCC Treatment

In certain HCC patients, immunotherapy reduces side effects, improves survival rates, and offers long-term cancer control. A wider range of patients may benefit from further study of immunotherapy in combination with existing treatments for HCC in its early and intermediate phases. PD-1/PD-L1, TMB, ctDNA, microsatellite stability, DNA mismatch repair, neutrophil/lymphocyte ratio, cytokines, and cellular peripheral immune response are being further studied to identify the most accurate marker for systemic treatment sequencing and selection, maximizing outcomes for HCC patients [245].

In HCC, cancer vaccines often fail to induce sustained, permanent immune responses due to the presence of immune checkpoints and a hepatic microenvironment that restricts immunity. Immune checkpoint inhibitors have significantly advanced the treatment of cancer. Nonetheless, numerous obstacles continue to impede the advancement of immunotherapy medications. CTLA-4, PD-1 and PD-L1, and lymphocyte activation gene 3 (LAG3) are the most thoroughly researched immune checkpoint molecules. The TME creates immunotolerant conditions through these pathways, making it difficult to trigger antitumor immune responses [246].

The goal of immune checkpoint therapy (ICT) is to enhance antitumor immune responses by blocking inhibitory signals of T-cell activation. ICT has provided long-term clinical benefits in a large population of patients with multiple tumor types, including cure in a subset of patients, since its successful introduction as a treatment for unresectable or metastatic melanoma in 2011. As a result, ICT’s clinical success transformed the field of cancer immunotherapy and made it a cornerstone of cancer treatment, alongside more conventional treatment modalities like radiation, chemotherapy, and surgery [247].

By inducing effector T-cell infiltration into tumors and immune checkpoint signals, cancer vaccinations may prepare patients for immune checkpoint inhibitor therapy. Clinical trials are now being conducted to explore the potential synergistic effects of an immune checkpoint inhibitor with a cancer vaccination in eliciting more potent antitumor immune responses [246].

Combination approaches integrating cancer vaccines with immune checkpoint inhibitors represent a promising approach for HCC treatment. Preclinical studies demonstrate that vaccines, including multi-peptide formulations, GM-CSF-secreting GVAX, and STINGVAX, can synergize with PD-1 or dual PD-1/CTA-4 blockade to enhance tumor-specific immune responses, overcome resistance, and improve antitumor efficacy [248]. Early clinical trials in pancreatic cancer suggest that vaccine-ICI combinations may reverse an immunosuppressive TME and induce lasting disease stabilization, although immune-related adverse events remain a consideration, particularly with CTLA-4 blockade [249]. Integrating ICIs with locoregional and targeted therapies enhances antitumor efficacy, offers a comprehensive strategy for HCC management, and provides a promising approach to broaden clinical benefits across diverse patient populations [250].

## 11. Translational Pipeline for Developing HCC Vaccines

The development of cancer vaccines follows a multistep translational pipeline that integrates basic immunology, tumor genomics, and clinical evaluation [251]. Initially, TAAs/TSAs such as AFP, GPC3, WT1, and hTERT are identified and validated through genomic and profiling of HCC tissues [252]. Finding the antigen’s epitopes is the next step after determining which antigen is best. Certain regions of an antigen are identified by the immune system as epitopes. Characterizing these important sections gives us the chance to create a vaccine that targets these epitopes. The vaccine candidate can then be created after the essential epitopes have been located and put together with the help of adjuvants and linkers. Finally, the assessment of vaccine design can be thoroughly done by using computational techniques [253].

Numerous in silico investigations can be carried out to examine the safety, effectiveness, toxicity, and interactions of drugs. To predict epitopes that may elicit T cell and B cell responses, in silico studies are conducted during the vaccine design process. The field of bioinformatics known as immunoinformatics examines the connection between anticipated epitopes and immune responses. Recent advancements in immunoinformatics have produced a number of tools for predicting antigenic responses and the epitopes recognized by B or T cells [254].

## 12. Challenges and Limitations in HCC Vaccine Development

The development of cancer vaccines for HCC faces several scientific, clinical, and practical obstacles, despite promising developments in cancer immunotherapy. These limitations hinder large-scale translation, trial comparability, patient eligibility, and overall vaccine potency [255]. Personalized neoantigen-targeting personalized cancer vaccines have generated strong attraction. However, key challenges, including accurate neoantigen identification, rapid, good manufacturing practice (GMP)-compliant manufacturing, and reliable assessment of clinically significant immune responses, continue to limit their widespread use. Experiences from clinical trials have highlighted critical design limitations and informed necessary improvements for next-generation platforms. While shared-antigen cancer vaccines can be mass-produced as ready-to-use formulations, each personalized neoantigen vaccine must be produced as an individual product, demanding rapid productions, scalable workflows, and strict cost control. To address these limitations, synthetic vaccine techniques that enable fast, economical production via a straightforward, reliable, invariant, and GMP-compliant procedure are crucial [13].

Vaccines against infectious diseases are highly effective, but their success in treating HCC remains limited. Early failures are due to reliance on TAAs. Neoantigen vaccines have a conceptual advantage because they target tumor-specific mutations absent from normal tissues, thereby inducing stronger, more specific T-cell responses with reduced risk of off-target toxicity. However, these vaccines require individualized tumor sequencing, intricate AI-driven epitope prediction, and expensive manufacturing. Although most clinical studies employ intramuscular or subcutaneous delivery, intra-tumoral vaccination may better reshape the suppressive TME and enhance local T-cell priming. Further clinical research is needed to determine whether this approach can enhance neoantigen vaccine performance in HCC [256].

Current regulatory and CMC frameworks have not been designed for individualized biologics, making consistent evaluation of safety and potency difficult. Therefore, developing platform-based, digitized manufacturing systems is essential to support the efficient, broader clinical applicability of personalized HCC vaccines [130].

Underlying clinical heterogeneity represents a significant hurdle in HCC vaccine development. Most of patients present with cirrhosis, MASLD, age, pre-existing inflammatory condition, or metabolic dysfunction such as type 2 diabetes. Age, gender, co-infections (e.g., HIV), pre-existing, and comorbidities further influence vaccine-induced immunity. These factors impair antigen presentation and T cell priming, limiting the efficacy of vaccines that rely on intact antigen processing machinery. Therefore, vaccination strategies must be evaluated not only in healthy individuals but also in populations with chronic liver disease and systemic inflammation [257].

Therapeutic vaccines for HCC have shown limited success primarily due to immune tolerance against self-derived TAAs, the inherently hepatic environment, and the profoundly immunosuppressive TME [13]. Early peptide and DC-based vaccines targeting antigens such as GPC3, AFP, and TERT often failed to induce strong CD8^+^ T-cell responses because central and peripheral tolerance limits high-avidity immunity against these TAAs [258]. Moreover, even when antigen-specific T cells were induced, their therapeutic potential was further compromised by the metabolic exhaustion, regulatory-cell enrichment, and checkpoint upregulation characteristics of the HCC TME, which collectively suppress effector function and limit the clinical impact of early vaccine strategies [259].

Moreover, in cirrhotic or chronically HBV/HCV-infected livers, continuous antigen exposure drives CD8^+^ T cells into a deeply exhausted state marked by high PD-1 expression and loss of effector function. This is linked to poor outcomes in HCC and is one that further undermines the impact of antigen-specific vaccines [260]. Importantly, recent advances demonstrate that personalized neoantigen vaccines, particularly combined with immune-checkpoint inhibitors, can induce robust CD8^+^ T-cell responses, enhance tumor infiltration and improve antitumor activity. These findings demonstrated that carefully optimized antigen selection, delivery platforms, adjuvant use, and combination strategies can partially overcome tolerance and TME-mediated immunosuppression, offering renewed hope for effective vaccine-based therapy in HCC [261].

## 13. Future Perspectives

The future of cancer vaccines is promising, driven by rapid advances in genomics, bioinformatics, and vaccine engineering that enable precise identification of tumor specific neoantigens and the development of highly personalized vaccines [262,263]. Personalized and combination therapy will play a major role in improving the efficacy of cancer vaccines [264]. Tumor heterogeneity can be addressed, and treatment precision can be increased by customizing vaccine formulations to each patient’s unique tumor features. Immune checkpoint inhibitors and adoptive cell treatments are examples of complementary immunotherapeutic approaches that can be integrated with various vaccine platforms to enhance antitumor immune responses and improve clinical outcomes [265]. While the advent of sophisticated antigen-delivery technologies, such as lipid nanoparticles and viral vectors, offers prospects to boost vaccine potency, specificity, and overall therapeutic efficiency, the identification of highly immunogenic TAAs or neoantigens remains crucial to vaccine success [266]. For cancer vaccine therapy to be optimized, predictive biomarkers and models are essential [267].

Future developments in cancer DNA vaccines will depend on improving and standardizing delivery methods, as current research varies greatly in dose, formulation, and DNA constructs, which limits reproducibility. Improved efficiency, scalability, and consistency in vaccine production are made possible by technological advancements in nanoparticle manufacturing, especially microfluidics and sophisticated mixing techniques. To improve immune responses and clinical efficacy and ultimately increase the therapeutic potential of DNA-based cancer immunotherapies, simultaneous optimization of delivery platforms, administration routes, injection sites, and devices will be crucial [268].

Predictive biomarkers and monitoring tools—including PD-1/PD-L1 expression, tumor mutational burden (TMB), circulating tumor DNA (ctDNA), microsatellite stability, DNA mismatch repair status, cytokine profiles, and peripheral immune cell responses—will be crucial for guiding patient selection, optimizing treatment sequencing, and enabling early detection of response or resistance. Establishing robust models for patient stratification and response monitoring can ensure more precise and effective therapeutic interventions [269,270].

Finally, significant cooperation between academia, business, and regulatory bodies is needed to expedite the development and approval of cancer vaccines for the treatment of HCC. To better understand the safety, effectiveness, and long-term benefits of cancer vaccines across diverse patient populations, this collaboration should include the development of more reliable preclinical models, the optimization of clinical trial designs, and the integration of empirical data [271].

## 14. Conclusions

HCC remains a global health challenge, with rising incidences and limited therapeutic option for advanced disease. Cancer vaccines offer a complementary, antigen-directed immunotherapeutic strategy capable of inducing tumor-specific cellular and humoral responses while establishing lasting immune memory. Advances in antigen discovery, neoantigen prediction, nucleic acid vaccine platforms, and nanoparticle-based delivery systems have substantially accelerated the discovery of new cancer vaccines. Strategies like surface modifications, self-amplifying mRNA constructs and tailored adjuvants can help overcome limitations like inadequate antigen delivery and restricted immunogenicity. However, tumor heterogeneity, the immunosuppressive TME and immune tolerance can reduce the efficacy of vaccines. Future progress will depend on refining antigen selection, optimizing vaccine delivery, and integrating vaccines with other immunotherapies to enhance antitumor efficacy. Ultimately, cancer vaccines are poised to become valuable components of multimodal strategies aimed at improving outcomes for HCC patients.

## Figures and Tables

**Figure 1 cimb-47-01056-f001:**
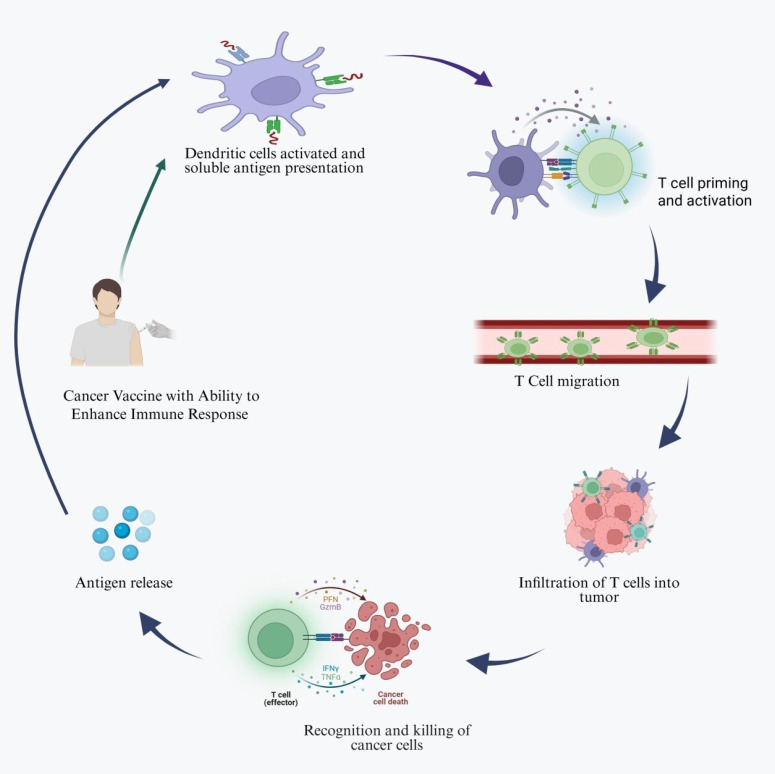
Schematic illustration of the generalized mechanism of cancer vaccines. Cancer vaccines activate the host immune system by delivering tumor-associated or tumor-specific antigens that are captured and processed by dendritic cells. Created in https://BioRender.com (accessed on 1 October 2025).

**Figure 2 cimb-47-01056-f002:**
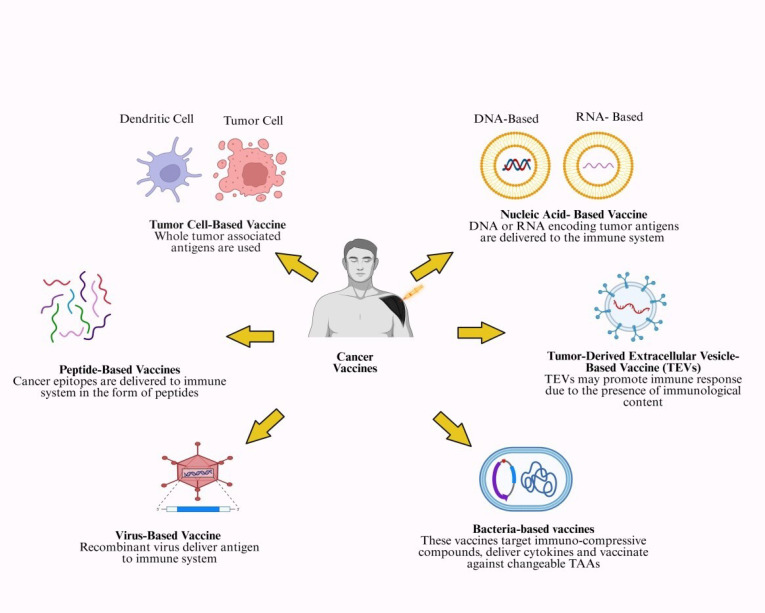
Various kinds of cancer vaccines being tested clinically. This figure summarizes types of cancer vaccines (tumor cell, peptide, nucleic acid, tumor-derived extracellular vesicles, bacteria-based and virus-based vaccines). Created in https://BioRender.com (accessed on 1 October 2025).

**Table 1 cimb-47-01056-t001:** Clinical evaluation of DNA-based cancer vaccines: targets, dosing regimens, and immunogenic outcomes.

Cancer Type	Target Antigen	Study Plan	Preclinical/Clinical Phase	Outcome Measures	Trial ID
Breast cancer	Personalized polypeptide DNA	4 mg vaccine at Day 1, 29 (±/1–7 days), and Day 57 (±7 days) with interval least 21 days interval between each dose, Intramuscular	1	Safety and Immunogenicity of personalized polyepitope DNA vaccine strategy	NCT02348320
Breast cancer	Mammaglobin-A antigen	4 mg vaccine on days 28, 56, and 84), intramuscular, Via TriGrid electroporation system	1	Safety, immune response, progression-free survival, overall survival, objective tumor response rate	NCT02204098
Melanoma	Mouse TYRP2 DNA	Intramuscularly at four escalating doses (500, 2000, 4000 or 8000 μg) at three-week intervals for six immunizations.	1	Safety and feasibility and any antitumor response generated after immunizations.	NCT00680589
Melanoma	gp75 DNA vaccine	1 mL, 2 mL or 4 mL every 3 weeks across 5 sessions	1	Safety and feasibility of intramuscular vaccination in patients with stage III or IV melanoma.	NCT00034554
Prostate cancer	Rhesus Prostate Specific Antigen (rhPSA)	Patients divided into 4 cohorts receiving 50 µg, 150 µg, 400 µg, or 1000 µg DNA/dose, administered intradermally with electroporation assistance	1 and 2	Feasibility and safety of escalating doses	NCT00859729
Non-Small cell Lung cancer	Semi-allogeneic human fibroblasts transfected with genomic tumor DNA	4 consecutive weekly injections, intradermally using a 1 mL syringe fitted with a 25-gauge needle.	1	Safety and feasibility, immune responses to the autologous tumor	NCT00793208
Liver cancer	Alpha fetoprotein (AFP) DNA and sargramostim (GM-CSF) plasmid DNA	Administrated intramuscularly on days 1, 30, and 60, followed by a booster dose on day 90 comprising AFP adenoviral vector via both intramuscular and intradermal routes.	1, 2	Safety and immunogenicity, overall survival	NCT00093548
Hepatocellular Carcinoma	Glypican-3 (GPC3)	DNA plasmid vaccine (NWRD06) administered by electroporation; dose escalation cohorts: 1 mg, 4 mg, 8 mg	1	Safety and immunogenicity	NCT06088459
Chronic Hepatitis Hepatitis C Infection Hepatocellular Carcinoma	INO-8000 (HCV antigen DNA) alone or co-administered with INO-9012 (interleukin [IL]-12 adjuvant DNA)	INO-8000 (HCV antigen DNA) alone or co-administered with INO-9012 (interleukin [IL]-12 adjuvant DNA)	1	Safety; HCV-specific CD4^+^/CD8^+^ T-cell responses	NCT02772003
Renal cancer	Human prostate-specific membrane antigen DNA vaccine, mouse prostate-specific membrane antigen DNA	6 intramuscular doses; alternating between mouse PSMA and human PSMA DNA vaccines in a sequential set of 3 dose each.	1	Safety and feasibility of vaccination, maximum tolerated dose, assess antitumor response	NCT00096629
Head and neck cancer	pNGVL-4a-CRT/E7 (detox) DNA	Intramuscular, 3 Doses (0.5, 1, 2, or 4 mg) using a TDS-IM device on days 1, 22, and 43 with 200 mg/m^2^ cyclophosphamide intravenously one day prior to each vaccination.	1	To evaluate adverse event associated with vaccine and immune response	NCT01493154
Glioblastoma	Neoantigen DNA	Vaccine once every 28 days for up to 6 doses, alongside Retifanlimab (500 mg every 28 days for up to 12 months), via electroporation-mediated intramuscular route	1	Safety, survival, overall survival, objective response rate	NCT05743595
Oral cancer	Dendritic Cells w/Tumor DNA	1 × 10^7^ DCs per dose injected intranodally or perinodally into lymph nodes distant from head and neck region	1	Safety and feasibility of immunization, immunological responses to the vaccine and/or antitumor immune responses	NCT00377247
Cervical Intraepithelial Neoplasia	GX-188E	1 mg or 4 mg of GX-188E per dose, at 0, 4 and 12 weeks intramuscularly using electroporation device.	2	Safety, tolerability, and finding the optimal dose of the vaccine	NCT02139267
Ovarian cancer	pUMVC3-hIGFBP-2 multi-epitope plasmid DNA	pUMVC3-hIGFBP-2 multi-epitope plasmid DNA vaccine intradermally once a month for 3 consecutive months.	1	Safety, immunogenicity, Disease-free survival, overall survival	NCT01322802

**Table 2 cimb-47-01056-t002:** Ongoing and completed clinical trials in mRNA cancer vaccines: design and outcomes.

Cancer Type	Antigen	Study Design	Phase	Outcome Measures	Trial ID
Breast cancer	Dendritic Cells Transfected with Survivin, hTERT and p53 mRNA	Combination of DC-based immunization with cyclophosphamide administration.	1	Toxicity, clinical tumor response, duration of tumor and immuno-response	NCT00978913
Melanoma	mRNA-nanoparticle (mRNA-NP)	3 intravenous doses of mRNA-NP vaccine (1 every 2 weeks), using 3 + 3 dose escalation design. mRNA dose range 0.00125–0.01 mg/kg.	1	Maximum tolerated dose, feasibility of treatment, overall response rate	NCT05264974
Prostate cancer	Dendritic Cells Loaded With mRNA from primary prostate cancer tissue	Intradermal vaccination with mRNA-loaded Dendritic Cells derived from patient tumor.	1 and 2	Time to treatment failure, safety and toxicity of vaccination. Evaluation of immunological response.	NCT01197625
Non-small cell lung cancer	Personalized MRNA Neoantigen	Vaccine given with adebrelimab as adjuvant therapy.	1	Safety, ability, immunogenicity, and preliminary efficacy	NCT06735508
Liver cancer	PD-1 mRNA LNP	Weekly doses (50–100 μg) for consecutive doses followed by a booster dose after 1 month.	1 and 2	Objective response rate, disease control rate, durable response rate, duration of response, response time	NCT07053072
Various tumors including hepatocellular carcinoma	Patient-specific tumor neoantigens (NCI-4650)	Intramuscular injections every 2 weeks for up to 4 doses (with option of second course ~4 weeks later).	1 and 2	Safety, immunogenicity (neoantigen-specific T-cell responses)	NCT03480152
HBV-positive Hepatocellular Carcinoma (waiting for liver transplantation)	HBV mRNA vaccine (HBV antigens)	Multiple mRNA vaccine doses (4 doses) during transplant waiting period.	1	Safety (AEs, DLTs), immunogenicity (antigen-specific T-cells), ORR, DCR, 1-year survival	NCT07077356
Advanced/metastatic Hepatocellular Carcinoma (after standard therapy failure)	PD-1 mRNA-LNP vaccine (PD-1 as immunogen)	Weekly injections × 4, then 5th dose after 1 month; dose escalation (low/med/high).	1 and 2	Safety, tolerability, immunogenicity, preliminary efficacy	NCT07053072
Advanced HBV-positive Hepatocellular Carcinoma (after failure of standard therapy)	HBV-mRNA vaccine	IM injections—dose escalation starting at 20 µg: weekly × 4 doses; 5th dose after one month.	1	Adverse events, objective response rate, progress-free survival, overall survival	NCT05738447
Advanced/relapsed/refractory Hepatocellular carcinoma (HCC)	mRNA-based personalized (neoantigen) vaccine—ABOR2014 (IPM511)	Intramuscular injection; 3 + 3 dose-escalation, two cycles (four injections per cycle).	Early phase	Safety, tolerability, immunokinetics/immunogenicity, preliminary efficacy	NCT05981066
Head and neck cancer	BNT113	Vaccine administrated in combination with pembrolizumab as adjuvant therapy.	2 and 3	Analysis of treatment-emergent adverse events, overall survival, progression-free survival	NCT04534205
Glioblastoma	Brain tumor stem cells mRNA-loaded dendritic cells	Escalating intradermal dose (2 × 10^6^–2 × 10^7^ DCs), given weekly for 3 doses, then monthly.	1	Humoral and cellular immune responses	NCT00890032
Intraepithelial Neoplasia and Cervical Cancer	NWRD09	NWRD09 administered by intramuscular injection.	NA	Assessment of immunogenicity, histopathological improvement, HPV viral clearance, objective response rate, progression-free survival, duration of response, disease control rate	NCT07092007
Ovarian cancer	8 W_ova1	Immunization with W_ova1 vaccine containing 3 ovarian cancer TAA RNA-LPX products.	1	Identification of patients exhibiting new or enhanced systemic immune responses to at any three vaccine antigens	NCT04163094
Advanced Digestive System Neoplasms	Personalized mRNA Tumor Vaccine Encoding Neoantigen	Subcutaneous administration of the personalized mRNA vaccine at least four times.	NA	Monitoring of treatment-related adverse events per CTCAE v4.0	NCT03468244
Advanced Gastric Cancer, Esophageal Cancer and Liver Cancer	Neoantigen tumor vaccine with or without PD-1/L1	Dose escalation with tumor vaccine alone followed by a combination with PD-1/PD-L1 inhibitor during dose expansion.	NA	Safety, objective response rate, progression-free survival, concentration of serum cytokine	NCT05192460

**Table 3 cimb-47-01056-t003:** Overview of clinical evaluation of peptide-based cancer vaccines across multiple tumor types.

Cancer Type	Antigen	Study Design	Preclinical/Clinical Phase	Outcomes	Trial ID
Breast cancer	9 Peptides from Her-2/neu, CEA, & CTA	Nine synthetic class I MHC-restricted peptides and a class II MHC-restricted tetanus helper peptide administrated intradermally and subcutaneously at days 1, 8, 15, 36, 43, and 50 using Montanide ISA-51	1	Evaluation of T-cell activation and infiltration into primary breast tumors	NCT00892567
Melanoma	4-peptide and 12-peptide melanoma	Vaccine administered over 6 weeks on 1, 8, 15, 29, 36, 43 days	2	Safety, immunogenicity, disease-free survival	NCT00938223
Prostate cancer	Peptide vaccine	Peptides administrated with escalating doses (100 mcg, 300 mcg, and 1 mg) combined with Poly IC-LC across 3 arms	NA	Monitoring adverse events of grade 3 or higher	NCT00694551
Non-small cell lung cancer	Neoantigen	Personalized neoantigen peptide vaccines delivered subcutaneously weekly for 12 weeks at 200 μg per peptide	1	Safety	NCT04397926
Renal cancer	Peptide	Intradermal peptide vaccine combined with granulocyte macrophage colony stimulating factor (GM-CSF)	1 and 2	Tolerability	NCT02429440
Liver cancer	Peptide	Custom peptide vaccine including autologous heat shock protein 70 and activated monocytes for patients with recurrent HCC post surgery	1	Safety, immunological response, progression-free survival, overall survival time	NCT05059821
Unresectable advanced Hepatocellular carcinoma	CD4-Th1 inducer cancer vaccine: UCPVax (telomerase-derived) + standard-of-care immunotherapy (anti-PD-L1 + anti-VEGF)	Subcutaneous UCPVax (0.5 mg, with adjuvant Montanide ISA-51) + IV Atezolizumab + Bevacizumab, per protocol schedule	2	Objective response rate, overall survival, progression-free survival, disease control rate	NCT05528952
Unresectable/recurrent/metastatic Hepatocellular carcinoma (HLA-A*2402)	VEGFR1 & VEGFR2 epitope peptides (anti-angiogenic peptide vaccine)	Antiangiogenic peptide vaccine for drugs include administration time frame, single group open label	1	Safety / toxicity	NCT01266707
Very early/early/intermediate Hepatocellular carcinoma (post-standard therapy)	IMA970A multi-peptide vaccine + CV8102 adjuvant	Single low-dose cyclophosphamide → nine intradermal vaccinations of IMA970A + CV8102	1 and 2	Safety/tolerability; antigen-specific CD8^+^ and CD4^+^ T-cell responses; immunologic parameters; survival/recurrence-free follow-up	NCT03203005
Advanced or metastatic cancer (various solid tumors, including hepatobiliary cancers)	Recombinant fowlpox-CEA (6D)/TRICOM vaccine ± Sargramostim (GM-CSF)	Vaccine therapy (viral-vector) with or without adjuvant (sargramostim), in patients with advanced/metastatic cancer	1	Safety/tolerability; immunologic response; possibly tumor response/survival (as part of exploratory endpoints)	NCT00028496
Head and neck cancer	Mutant p53 peptide pulsed dendritic cell	Adjuvant p53 Peptide-Loaded DC-Based Treatment	1	Toxicity, immunological response rate, biologic response rate	NCT00404339
Gastric cancer	Peptides emulsified with montanide ISA51	Intracutaneous LY6K-177 peptide (1 mg in 1 mL sterile saline) into the inguinal region	1	Assessment of safety and efficacy	NCT00845611
Colorectal and pancreatic cancer	KRAS peptide	Vaccination on cycle 1 (days 1, 8, 15), R-Cycle 2 Day 1 (R-C2D1) with booster doses every 28 days in later cycles D: up to 1.8 mg peptide + 0.5 mg Poly-ICLC	1	Evaluation of toxicity, CD8 and CD4 T-cell responses, disease-free survival, objective response rate, interferon (IFN)-γ-producing T cells	NCT04117087
Pancreatic cancer	Neoantigen peptide	Intramuscular injections on days 1, 4, 8, 15, 22, 50, and 78 starting about 1 month post-surgery or roughly 1 week after Chemotherapy	1	Assessment of safety, immunogenicity	NCT05111353

**Table 4 cimb-47-01056-t004:** Clinical evaluation of cell-based cancer vaccines across multiple tumor types: study designs, interventions, and outcomes.

Cancer Type	Interventions	Study Design	Preclinical/Clinical Phase	Outcomes	Trial ID
Breast cancer	Dendritic cell	Intravenous administration of 20–30 million cells per injection, at least 3 times	1	Cytotoxicity, peripheral blood parameters, immune indicator, immunosuppressive content	NCT03113019
Melanoma	Bystander-Based Autologous Tumor Cell	Three intradermal injections on days 1, 29, and 57 spaced 28 days apart	2	Partial response, serious adverse events, stable disease rate, time to progression, overall survival	NCT00101166
Prostate cancer	Prostate cancer antigen	Autologous dendritic cells pulsed with prostate cancer antigen and KLH	1 and 2	PSA levels, time to progression, overall survival, immune and clinical response	NCT01171729
Non-Small Cell Lung Cancer	Personalized tumor neoantigen	Four consecutive cycles, each cycle up to four subcutaneous injections at four limb sites, combined with pembrolizumab	2	FRAME-001-specific immune responses, adverse events, tumor response, survival	NCT04998474
Hepatocellular Carcinoma	Tumor neoantigen	Seven DC vaccination sessions combined with microwave ablation, first injection 30 days post ablation	1	Safety, immunogenicity, progression-free survival	NCT03674073
Hepatocellular carcinoma	COMBIG-DC (ilixadencel)—allogeneic dendritic-cell based therapeutic vaccine (tumor-pulsed DCs)	Intra-tumoral injection COMBIG-DC (allogeneic dendritic cells) Cancer Vaccine 3 vaccinations: 5, 10 or 20 million cells per injection	1	Safety and tolerability	NCT01974661
Renal cancer	Autologous renal cell vaccine based on DNP-modified cells	Combined with sunitinib treatments	1 and 2	Immunological response, progression-free survival, dermatologic and allergic reaction	NCT00890110
Head and neck squamous cell carcinoma	MVX-ONCO-1	Weekly vaccination for four weeks, followed by two additional doses every two weeks apart, each dose contains Two macrocapsules with irradiated autologous tumor cells up each dose.	NA	Overall survival, time to subsequent therapy, duration of response, object response rate, disease control rate	NCT02999646
KK-LC-1 positive epithelial cancers	KK-LC-1 TCR T cells + aldesleukin	Non-myeloablative, lymphocyte depleting preparative regimen, followed by escalating doses of TCR T cells	1	Tolerated dose	NCT05035407
Colorectal cancer	Autologous tumor material	4 immunizations at weeks 0, 4, 8, and 12, using 2 million dendritic cells. Intradermal, intranodal, or intralymphatic injections	1	Feasibility, safety, immunity results	NCT00558051
Pancreatic cancer	Multi-antigen	Combination therapy with multiple agents including ALT-803, ETBX-011, GI-4000, chemotherapeutics and immunotherapies	1 and 2	Treatment-emergent adverse events, serious adverse event	NCT03387098

**Table 6 cimb-47-01056-t006:** Clinical evaluation of peptide and dendritic cell-based vaccines targeting AFP, GPC3, and hTERT in hepatocellular carcinoma.

Vaccine	Number of Patients	Characteristics	Immune Response	Conclusion of Study	Reference
Immunodominant HLA-A*0201-restricted peptides	6	AFP-positive HCC	T-cell response was generated	Even high quantities of AFP, human T cell can still identify AFP when presented by MHC class I	[220]
AFP-derived peptides (AFP357 and AFP403)	15	Patients with advanced HCC, and characterized induction of AFP-specific T-cell receptors (TCRs)	Expressed a highly functional TCR	No adverse events and produced T cells with receptors	[218]
HepaVac-101	82	Patients with very early- to intermediate-stage HCCs were enrolled and screened for suitable HLA haplotypes	Induced HLA class I TAA responses in 37% and class II TAA responses in 53% patients	HepaVac-101 showed safety and immunogenicity, supporting further clinical investigation	[221]
AFP peptide pulsed DC vaccine	10 treated/16 enrolled	HLA-A*0201 patients with AFP-positive HCC patients	In 6 patients, vaccination expanded AFP-specific T cells and boosted IFNℽ responses to at least one peptide	Human T cells can mount responses to AFP despite high circulating levels when stimulated with AFP-pulsed DCs	[222]
Hepcortespenlisimut-L	75	Patients with advanced HCC	CD4^+^ and CD8^+^ T cells activated, increased IFN-ℽ, reduced TNF-α	Hepcortespenlisimut-L is safe, effective, and fast-acting immunotherapy for HCC	[223]
Neo-DCVac-02	32	Patients had high-risk HCC after radical surgery	Increased antigen-specific T-cell activity, sustained cellular responses, Increases in T-cell activation/co-stimulation	Neo-DCVac-02 is safe, well-tolerated, and elicits durable antigen specific immune response, potentially delaying HCC recurrence	[224]
GPC3 peptide	33	Patients with advanced HCC	A GPC3-specific CTL response was induced	GPC3 peptide vaccine was safe, induced immune and antitumor response, and CTL frequency predicted overall survival in HCC	[219]
GPC3 derived peptide	41	Patients with initial HCC who had undergone surgery or radiofrequency ablation (RFA)	Strong antigen-specific T-cell responses in PBMCs	GPC3 expression in primary tumor may serve as a biomarker for evaluating the GPC3 peptide vaccine in future trials	[225]
A telomerase peptide (GV1001)	40	Patients with advanced HCC	GV1001 reduced regulatory T cells but did not elicit detectable vaccine-specific immune response	Low-dose cyclophosphamide plus GV1001 did not show antitumor activity	[226]

**Table 7 cimb-47-01056-t007:** Representative ongoing or completed clinical trials of cancer vaccines in HCC, highlighting antigens, vaccine platforms, phases, and outcomes.

Vaccine Platform	Target Antigens	Study Design	Clinical Phase	Key Outcomes	Trial ID
COMBIG-DC (allogeneic dendritic cells)	AFP and hTERT	3 vaccines of 5, 10 or 20 million cells per dose	1	Adverse events, safety, tolerability, systemic inflammatory response	NCT01974661
Anticancer vaccine (CRCL-AlloVax)	AFP	Priming intradermal AlloStim on Days 0–10, vaccination intradermal AlloStim + CRCL on Days 14–24, Activation intravenous AlloStim on Day 28, booster monthly intradermal CRCL on Day 56	2	Survival, safety	NCT02409524
Cancer stem cell-loaded DC vaccines	Specific antigen in metastatic adenocarcinoma	Randomized, controlled with 4 arms–placebo (non-cancer stem cell vaccine) and three experimental groups receiving low, medium or high dose CSC Vaccines	1 and 2	Rate of adverse event, immune response rate,	NCT02089919
AFP gene hepatocellular carcinoma vaccine	AFP	Three biweekly intradermal four HLA-A*0201-binding AFP-derived peptides (100 μg each) emulsified in 2 mL Montanide ISA-51	1 and 2½	Safety and tolerability, immune response	NCT00005629
Peptide-based vaccine with IL-2 or GM-CSF	Peptide antigen	3 arm trial using ras peptide vaccine ± IL-2 and/or GM-CSF given subcutaneously in repeated cycle	2	Effectiveness, safety, toxicity	NCT00019331
Dendritic Cells Pulsed With Four AFP Peptides	AFP	3 intradermal biweekly AFP doses to groups of 3 patients with escalation after 30 days of safety observation	1 and 2	Progression-free survival, clinical responses	NCT00022334
AFP + GM-CSF Plasmid Prime And AFP Adenoviral Vector Boost	AFP	AFP DNA+ GM-CSF-intramuscular vaccines (1, 30, 60) followed by AFP adenoviral boost (day 90); dose escalation to MTD, follow-up monthly, up to 6 months	1 and 2	Safety, immunogenicity	NCT00093548

## Data Availability

No new data were created or analyzed in this study. Data sharing is not applicable to this article.

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
