# Peer review of "Cancer Vaccines: Molecular Mechanisms, Clinical Progress, and Combination Immunotherapies with a Focus on Hepatocellular Carcinoma"

_cimb, 2025, doi:10.3390/cimb47121056_

Round 1
Reviewer 1 Report
Comments and Suggestions for Authors
The review article titled “Cancer Vaccines in Hepatocellular Carcinoma: Molecular Mechanisms, Clinical Progress, and Combination Immunotherapies" is a comprehensive overview of cancer vaccine technologies applied to HCC. The authors have done a good job compiling substantial clinical trial data and extensive information on immunotherapy mechanisms. The article is clinically relevant and timely.
However, the manuscript has major concerns: there is a significant mismatch between the focus implied by the title and the actual content, the organization of sections hampers flow, and there is considerable redundancy. Substantial revision to address the points below will significantly improve the manuscript and make it more focused and valuable.
- The title and content do not match. The stated focus is HCC, but roughly half (or more) of the manuscript discusses general cancer vaccine principles applicable to many malignancies. The manuscript should either be revised to concentrate more on HCC, or the title should be broadened to reflect the wider scope (for example: “Cancer Vaccines: Molecular Mechanisms, Clinical Progress, and Combination Immunotherapies with a Focus on Hepatocellular Carcinoma”).
- The manuscript lacks a clear structural organization, which disrupts flow and readability. The authors should adopt a more logical sequence. The clinical applications in HCC—the most relevant section—appears too late. Consider restructuring as: Introduction → Vaccine Types → TME/Resistance → Key Components → HCC-Specific Considerations → Clinical Applications in HCC → Combinations → Future Directions → Conclusion.
- The TME section should be substantially revised to emphasize the immunosuppressive, fibrotic nature of the HCC tumor microenvironment.
- There are redundant materials that reduce the manuscript’s quality and should be removed. Examples include: lines 134–178 (repeated cancer vaccine introduction), lines 463–485 and 898–920 (repeated immune activation mechanisms), and lines 727–750 and 1193–1234 (repeated nanoparticle delivery discussion).
- Sections 5 and 6.1 should be combined into a single section on antigen selection.
- The concept of the immunosuppressive TME is repeated throughout the manuscript and should be consolidated.
- Section 9 lacks HCC-specific analysis and should be expanded to include HCC-specific tumor antigens (e.g., GPC3, AFP, hTERT) with molecular characterization.
- It is important to discuss, in the context of HCC, why certain vaccine approaches succeeded while others failed.
- If the authors choose to focus the manuscript specifically on HCC as suggested above, Tables 1–4 should be removed or consolidated to emphasize HCC-focused data.
- Add a section on challenges and limitations in HCC vaccine development that addresses patient selection, cost, complexity of development, and standardization.
- Some sections are only loosely relevant to HCC—for example, bacteria-based vaccines have minimal application in HCC—and should be removed or condensed.
- The delivery details section can be reduced by roughly half.
- There are numerous grammatical errors, phrasing problems, and typographical mistakes. The manuscript requires careful revision by a language expert. There are also citation issues (e.g., lines 204 and 313) where citations appear incomplete with author names in the text.
The English language needs significant improvement.
Author Response
The review article titled “Cancer Vaccines in Hepatocellular Carcinoma: Molecular Mechanisms, Clinical Progress, and Combination Immunotherapies" is a comprehensive overview of cancer vaccine technologies applied to HCC. The authors have done a good job compiling substantial clinical trial data and extensive information on immunotherapy mechanisms. The article is clinically relevant and timely.
However, the manuscript has major concerns: there is a significant mismatch between the focus implied by the title and the actual content, the organization of sections hampers flow, and there is considerable redundancy. Substantial revision to address the points below will significantly improve the manuscript and make it more focused and valuable.1. Title–content mismatch; manuscript is too broad and not sufficiently HCC-focused.
Response. We would like to express our sincere gratitude to our respected reviewers for their thoughtful, detailed, and highly constructive comments. Your insights have greatly strengthened the scientific depth, clarity, and overall quality of our manuscript.
Comment 1: The title and content do not match. The stated focus is HCC, but roughly half (or more) of the manuscript discusses general cancer vaccine principles applicable to many malignancies. The manuscript should either be revised to concentrate more on HCC, or the title should be broadened to reflect the wider scope (for example: “Cancer Vaccines: Molecular Mechanisms, Clinical Progress, and Combination Immunotherapies with a Focus on Hepatocellular Carcinoma”).
Response: We sincerely thank the reviewer for this important observation. We agree that substantial portions of the manuscript introduce general cancer vaccine concepts to build a mechanistic foundation. However, we understand that this may give the impression of a broader scope than the title originally suggested. Therefore, as recommended, we have revised the title to:
“Cancer Vaccines: Molecular Mechanisms, Clinical Progress, and Combination Immunotherapies with a Focus on Hepatocellular Carcinoma.”
This more accurately reflects the structure and emphasis of the review.
Comment 2: The manuscript lacks a clear structural organization, which disrupts flow and readability. The authors should adopt a more logical sequence. The clinical applications in HCC—the most relevant section—appears too late. Consider restructuring as: Introduction → Vaccine Types → TME/Resistance → Key Components → HCC-Specific Considerations → Clinical Applications in HCC → Combinations → Future Directions → Conclusion.
Response: We sincerely appreciate this suggestion. We have fully reorganized the manuscript according to the recommended structure to enhance readability and logical progression.
Comment 3: The TME section should be substantially revised to emphasize the immunosuppressive, fibrotic nature of the HCC tumor microenvironment.
Response: Thank you for this valuable comment. We have revised the TME section to more clearly emphasize the fibrotic, immunosuppressive characteristics unique to the HCC microenvironment.
Comment 4. There are redundant materials that reduce the manuscript’s quality and should be removed. Examples include: lines 134–178 (repeated cancer vaccine introduction), lines 463–485 and 898–920 (repeated immune activation mechanisms), and lines 727–750 and 1193–1234 (repeated nanoparticle delivery discussion)..
Response. We greatly appreciate the reviewer’s careful reading. The Introduction provides a general overview, whereas lines 134–178 (Now in section 2) present a more detailed discussion. Therefore, this section cannot be removed, as it adds necessary elaboration beyond the general summary. The content of Line 898–920 (Now in section 5.2 in revised manuscript) and 1193–1234 (Now in section 7.1. in revised manuscript) are revised to reduce redundancy.
Comment 5. Sections 5 and 6.1 should be combined into a single section on antigen selection..
Response: We thank the reviewer for this logical suggestion. We merged the sections (section 5.1. now) and highlighted.
Comment 6. The concept of the immunosuppressive TME is repeated throughout the manuscript and should be consolidated.
Response: We agree with the reviewer’s concern and have reduced redundancy related to the immunosuppressive TME throughout the manuscript. However, some context has been retained in specific sections where it is essential for maintaining clarity and supporting section-specific discussions.
Comment 7. Section 9 lacks HCC-specific analysis and should be expanded to include HCC-specific tumor antigens (e.g., GPC3, AFP, hTERT) with molecular characterization.
Response: We appreciate the reviewer’s suggestion. Section 9 (now section 8) has been revised accordingly, and the recommended HCC-specific tumor antigens (GPC3, AFP, hTERT), along with their molecular characteristics, have been incorporated to strengthen the section.
Comment 8. It is important to discuss, in the context of HCC, why certain vaccine approaches succeeded while others failed.
Response: We sincerely appreciate this insightful recommendation. The revised manuscript now includes a dedicated discussion (section 12 Challenges and limitations in HCC vaccine development: Paragraph 2) on factors influencing success or failure of vaccine approaches in HCC.
Comment 9. Tables 1–4 should be removed or condensed for HCC-specific focus.
Response: We deeply appreciate this thoughtful comment. While the revised manuscript is more HCC-centered, our review aims to provide a mechanistic and translational understanding of cancer vaccines with a focus on HCC, not an exclusively HCC-only review. We maintain that Tables 1–4 provide essential context on cancer vaccine platforms, delivery systems, and clinical progress that underpin HCC vaccine development. However, to address the reviewer’s concern: We have updated each table to clearly highlight the HCC-relevant clinical data. We note that very few completed or data-reporting HCC vaccine trials are currently available. Therefore, entirely HCC-exclusive tables are not feasible at this stage. We sincerely apologize for any earlier lack of clarity and trust the revised tables now more effectively support the manuscript’s HCC-focused narrative.
Comment 10. Add a section on challenges and limitations in HCC vaccine development that addresses patient selection, cost, complexity of development, and standardization.
Response: We are grateful for this recommendation. We have added a comprehensive section (section 12 Challenges and limitations in HCC vaccine development) addressing patient selection, cost, complexity, standardization challenges, and regulatory considerations.
Comment 11. Some sections are only loosely relevant to HCC—for example, bacteria-based vaccines have minimal application in HCC—and should be removed or condensed.
Response: We appreciate the reviewer’s concern. Although bacteria-based vaccines have limited direct application in HCC, this section is retained because the review covers broad cancer vaccine mechanisms with a focus on HCC, not exclusively HCC-specific platforms. Bacterial vectors provide important conceptual and mechanistic insights that are highly relevant to overcoming immune tolerance in HCC. Including this section ensures a comprehensive discussion of emerging vaccine technologies that may influence future HCC vaccine development.
Comment 12. The delivery details section can be reduced by roughly half.
Response: We thank our respected reviewer for this comment. Detailed descriptions of cancer vaccine delivery approaches are essential because delivery directly affects antigen protection, cellular uptake, immune activation, and overall vaccine efficacy. As different platforms operate through distinct mechanisms, this information is crucial for interpreting vaccine outcomes and comparing strategies. Therefore, we have retained this section without reducing its content.
Comment 13. There are numerous grammatical errors, phrasing problems, and typographical mistakes. The manuscript requires careful revision by a language expert. There are also citation issues (e.g., lines 204 and 313) where citations appear incomplete with author names in the text.
Response: We sincerely thank the reviewer for this observation. The entire manuscript has undergone thorough language editing, correction of hyphenation issues, removal of incomplete sentences, and fixing of citation inconsistencies.
Reviewer 2 Report
Comments and Suggestions for Authors
The article is a broad ranging review covering recent clinical trials across multiple cancer vaccine modalities, including viruses, genetically-engineered bacteria, transfection of plasmid DNA, and other approaches. They focus on liver cancer (Hepatocellular Carcinoma), but also consider vaccine trials in other cancers. My one area for improvement is that the authors largely provided a list of recent trials and outcomes, with little or no in-depth analysis of specific trials. However, given the sheer number of papers reviewed, this is not feasible for every report, so perhaps going into greater critical evaluation for 1-2 key trials in each section may be reasonable. However, the paper is already fairly long, so this may raise other issues.
The authors have generated a timely review of cancer vaccination approaches, including brief overviews of relevant clinical trials, which will be helpful to scientists working in this field. My one concern is the numerous language issues (see below)
Comments on the Quality of English LanguageThe manuscript is in need of editing by a native English language speaker, as well as extensive proof-reading. Specific points include the often bizarre hyphenation usage (could this have been an issue from shifting formats? it needs correction, in any event), incomplete sentences (line 736 and line 793 for example - this is not exhaustive), and what appears to be a repeated paragraph (lines 959-963 repeats 964-968).
Author Response
Comment 1. The article is a broad ranging review covering recent clinical trials across multiple cancer vaccine modalities, including viruses, genetically-engineered bacteria, transfection of plasmid DNA, and other approaches. They focus on liver cancer (Hepatocellular Carcinoma), but also consider vaccine trials in other cancers. My one area for improvement is that the authors largely provided a list of recent trials and outcomes, with little or no in-depth analysis of specific trials. However, given the sheer number of papers reviewed, this is not feasible for every report, so perhaps going into greater critical evaluation for 1-2 key trials in each section may be reasonable. However, the paper is already fairly long, so this may raise other issues.
Response. We thank our respected reviewer for this valuable suggestion. We agree that deeper critical analysis of selected trials would strengthen the review; however, given the large number of studies included and the manuscript’s considerable length, providing detailed evaluations for additional trials is not feasible. The manuscript has already undergone extensive revisions, including updates suggested by other reviewers, which constrained further additions in this area. We sincerely appreciate the reviewer’s insight and apologize for not incorporating this additional change.
Comment 2. The authors have generated a timely review of cancer vaccination approaches, including brief overviews of relevant clinical trials, which will be helpful to scientists working in this field. My one concern is the numerous language issues (see below)
Response. We sincerely thank our respected reviewer for their positive assessment of the relevance and usefulness of our review. We fully acknowledge the language issues noted and have carefully revised the entire manuscript for grammar, clarity, and readability. We appreciate the reviewer’s careful reading and constructive feedback.
Comments on the Quality of English Language
Comment 3. The manuscript is in need of editing by a native English language speaker, as well as extensive proof-reading. Specific points include the often bizarre hyphenation usage (could this have been an issue from shifting formats? it needs correction, in any event),
Response. We sincerely thank the reviewer for the careful evaluation. We fully acknowledge the language issues noted and have thoroughly revised the manuscript to improve grammar, clarity, and readability. All incorrect or inconsistent hyphenation has been carefully corrected. We greatly appreciate the reviewer’s constructive feedback and attention to detail.
Comment 4. Incomplete sentences (line 736 and line 793 for example - this is not exhaustive).
Response. We sincerely thank the reviewer for the careful evaluation. We completed the sentences.
Overall, genetic engineering allows selective, large and controlled levels of target molecules to boost vaccine efficacy
For malignancies linked to viruses, such as cervical or oropharyngeal cancer caused by the human papillomavirus (HPV), Merkel cell carcinoma (MCC) caused by the Merkel cell polyomavirus (MCPyV), and head and neck cancers caused by the Epstein-Barr virus (EBV), any epitopes come from open reading frame [133].
Comment 5. what appears to be a repeated paragraph (lines 959-963 repeats 964-968).
Response- We sincerely thank the reviewer for this observation. We revised the content and removed repeated content.
Round 2
Reviewer 1 Report
Comments and Suggestions for Authors
The authors have revised the manuscript to a significant extent and have addressed several of the previously raised concerns. However, a few issues still require attention:
- Section 12 – “Challenges and limitations in HCC vaccine development”
The first and third paragraphs contain overlapping content. The authors should revise this section and improve clarity. - Section 13 – Future Perspectives
This section focuses almost exclusively on DNA vaccines. The discussion does not reflect the broader concept and current understanding of cancer vaccine platforms. In addition, the authors have not provided a future perspective for HCC vaccine development. A more comprehensive and balanced outlook is needed. - The earlier comment regarding the overly detailed description of vaccine delivery methods has not been fully addressed. While I agree that delivery strategies are relevant to the review, the current level of detail is excessive. As suggested previously, this content should be reduced by approximately half. Adding unnecessary length does not enhance the quality of the manuscript; instead, concise, meaningful, and focused information will strengthen it.
The English language needs improvement.
Author Response
- Section 12 – “Challenges and limitations in HCC vaccine development”
The first and third paragraphs contain overlapping content. The authors should revise this section and improve clarity.
Response- We thank the reviewer for highlighting the overlap between the first and third paragraphs of this section. To improve clarity and avoid redundancy, overlapping content has been removed and the section has been streamlined. Paragraph 3 has been substantially revised, with clearer differentiation of key challenges and improved logical flow.
2. Section 13 – Future Perspectives
This section focuses almost exclusively on DNA vaccines. The discussion does not reflect the broader concept and current understanding of cancer vaccine platforms. In addition, the authors have not provided a future perspective for HCC vaccine development. A more comprehensive and balanced outlook is needed.
Response- We appreciate the reviewer’s insightful comment regarding the scope of this section. In response, Section 13 has been thoroughly revised to present a more comprehensive and balanced outlook. This revision aligns the section more closely with current advances and future directions in cancer vaccine research.
- The earlier comment regarding the overly detailed description of vaccine delivery methods has not been fully addressed. While I agree that delivery strategies are relevant to the review, the current level of detail is excessive. As suggested previously, this content should be reduced by approximately half. Adding unnecessary length does not enhance the quality of the manuscript; instead, concise, meaningful, and focused information will strengthen it.
Response- We acknowledge the reviewer’s concern regarding the excessive detail in the description of vaccine delivery approaches. In response, the delivery-related content has been reduced by approximately half, with an emphasis on key concepts rather than procedural detail. Redundant technical descriptions were removed, and the remaining text was refined to provide concise, focused, and meaningful information that supports the overall narrative without unnecessary expansion.